# Psychological Health and Drugs: Data-Driven Discovery of Causes, Treatments, Effects, and Abuses

**DOI:** 10.3390/toxics11030287

**Published:** 2023-03-20

**Authors:** Sarah Alswedani, Rashid Mehmood, Iyad Katib, Saleh M. Altowaijri

**Affiliations:** 1Department of Computer Science, Faculty of Computing and Information Technology, King Abdulaziz University, Jeddah 21589, Saudi Arabia; 2High-Performance Computing Center, King Abdulaziz University, Jeddah 21589, Saudi Arabia; 3Department of Information Systems, Faculty of Computing and Information Technology, Northern Border University, Rafha 91911, Saudi Arabia

**Keywords:** psychological health, drugs, Twitter, machine learning, big data, drug abuse

## Abstract

Mental health issues can have significant impacts on individuals and communities and hence on social sustainability. There are several challenges facing mental health treatment; however, more important is to remove the root causes of mental illnesses because doing so can help prevent mental health problems from occurring or recurring. This requires a holistic approach to understanding mental health issues that are missing from the existing research. Mental health should be understood in the context of social and environmental factors. More research and awareness are needed, as well as interventions to address root causes. The effectiveness and risks of medications should also be studied. This paper proposes a big data and machine learning-based approach for the automatic discovery of parameters related to mental health from Twitter data. The parameters are discovered from three different perspectives: Drugs and Treatments, Causes and Effects, and Drug Abuse. We used Twitter to gather 1,048,575 tweets in Arabic about psychological health in Saudi Arabia. We built a big data machine learning software tool for this work. A total of 52 parameters were discovered for all three perspectives. We defined six macro-parameters (Diseases and Disorders, Individual Factors, Social and Economic Factors, Treatment Options, Treatment Limitations, and Drug Abuse) to aggregate related parameters. We provide a comprehensive account of mental health, causes, medicines and treatments, mental health and drug effects, and drug abuse, as seen on Twitter, discussed by the public and health professionals. Moreover, we identify their associations with different drugs. The work will open new directions for a social media-based identification of drug use and abuse for mental health, as well as other micro and macro factors related to mental health. The methodology can be extended to other diseases and provides a potential for discovering evidence for forensics toxicology from social and digital media.

## 1. Introduction

Several factors are contributing globally to declining social sustainability including people’s health, economic issues, global events such as the COVID-19 pandemic and environmental disasters, and increased social division and polarization [1]. These factors have caused negative impacts on the well-being and future prospects of our societies, leading to declining social sustainability. Social sustainability is closely linked to economic and environmental sustainability, as the economic conditions of a society and the state of the natural environment can both have major impacts on the well-being of its members. In order to address the risk of declining social sustainability, it is important to act to address the root causes of these issues.

Mental health is related to social sustainability because it is an important aspect of overall health and well-being, and mental health issues can have significant impacts on individuals and communities. Mental health issues such as depression and anxiety can lead to decreased productivity, absenteeism, suicides, and other negative impacts on social and economic well-being. For example, according to the World Health Organization (WHO), there is a suicide every 40 s, totaling more than 700,000 per year. This high rate of suicide highlights the deteriorating social conditions around the world [2].

Moreover, addiction is often related to mental health in that it can be a symptom of, or a response to, underlying mental health issues. For example, people may turn to substances or behaviors such as drugs, smoking, alcohol, gambling, or internet use as a way to cope with mental health issues such as depression, anxiety, or stress. However, addiction can also contribute to, or exacerbate, mental health problems, as the use of substances or engagement in certain behaviors can have negative impacts on mental well-being [3]. The Centers for Disease Control and Prevention (CDC) reports that cigarette smoking results in more than 480,000 deaths annually in the United States, with over 40,000 of these deaths attributed to second-hand smoke exposure. Additionally, over 16 million Americans have experienced severe health issues due to smoking [4]. The National Survey on Drug Use and Health (NSDUH) also reports that more than 19.5 million Americans over the age of 12 struggle with substance use disorders [5].

There are several challenges facing mental health treatment, including a lack of access to care, stigma, a shortage of mental health professionals, limited treatment options, co-occurring disorders, and a lack of integration with physical health care. These challenges can make it difficult for people to receive the mental health treatment they need, which can have negative impacts on their well-being and overall quality of life. Addressing these challenges is important for promoting mental health and improving the well-being of individuals and communities.

However, more important is to remove the root causes of mental illnesses because doing so can help prevent mental health problems from occurring or recurring, improve the effectiveness of treatment, and reduce the need for ongoing care. Root causes of mental health issues can include trauma, genetics, environmental factors, and physical health issues. A multifaceted approach that addresses social, economic, and environmental factors as well as individual needs is needed to remove the root causes of mental health issues effectively.

There is a significant body of research on the relationship between physical and psychological health. Studies have explored the connection between mental stress and physical diseases such as cancer, lung disease, and kidney disease [6,7,8], as well as the impact of physical conditions such as obesity and smoking on psychological health [9,10,11]. There is also research on specific psychological disorders, including depression, anxiety, stress, and post-traumatic stress disorder (PTSD) [12]. In the education field, there is research on the prevalence of psychological illnesses among students and academics and the impact of teachers’ mental health on students’ achievement [13]. The COVID-19 pandemic has also led to research on the effects of the pandemic on psychological health, including the spread of depression, anxiety, and stress among the general population as well as the psychological impact of quarantine and social distancing measures [14,15].

A holistic approach to understanding mental health issues is missing from the existing research. What is needed is to understand mental health and illnesses in the context of socio-economic and environmental contexts, create awareness for the people of the causes and effects of mental illnesses, and develop interventions to bring social behaviors, lifestyle, and root cause changes.

This paper proposes a big data and machine learning-based approach for the automatic discovery of parameters (or factors) related to mental health (or psychological health) from Twitter data. The parameters are discovered from three different perspectives Drugs and Treatments, Causes and Effects, and Drug Abuse. Moreover, we automatically discover associations between the parameters and drugs. We gathered from Twitter 1,048,575 tweets in Arabic about psychological health in Saudi Arabia during the month of October 2022. The tweets were retrieved using various keywords and hashtags related to mental health. We built a machine learning software tool for this work (see Section 3 for details). A total of 52 parameters were discovered for all three perspectives. We defined six macro-parameters to aggregate related parameters. We provide a comprehensive account of mental health, causes, medicines and treatments, mental health and drug effects, and drug abuse, as seen on Twitter, discussed by the public and health professionals.

## 2. State-of-the-Art

This section provides a review of the works related to our paper. We intentionally keep it short here. An extended version of this paper including the literature review has been made available as a preprint [16].

A good part of the research on psychological health has investigated the relationship between psychological illnesses and chronic physical diseases (e.g., cancer, lung, and kidney diseases) using different data sources. See for instance [6,7,8,17]. Studies have also been conducted on investigating the connection between obesity and mental health [9,10]. Some studies have looked into the relationship between smoking and psychological diseases [11,18]. Many works have investigated psychological diseases with a focus on specific factors or disorders such as depression, anxiety, stress, and post-traumatic stress disorder (PTSD) [12,19,20,21,22,23]. Researchers have also investigated the prevalence of psychological illnesses in students and academics [13,24,25,26,27,28,29,30]. Researchers have also explored the effects of COVID-19 on psychological health [14,15,31]. Several works studied the psychological effects of the COVID-19 pandemic on students and the education sector [32,33,34,35,36,37,38].

Discussing works that used machine-learning methods in studying topics related to mental health, for instance, Iram et al. [39] utilized random forests algorithm to distinguish between linguistic styles, detect depressive and non-depressive contents, and identify the degree of severity among contents on social media. Islam et al. [40] used various ML algorithms such as the Decision Tree classifier, SVM, and KNN for depression detection on Facebook. They examined four forms of factors of depression including the emotional, temporal, and linguistic style. Wang et al. [41] used sentiment analysis models for detecting depression in micro-blogs. Sentiment analysis employs natural language processing (NLP) techniques to discern and extract subjective information from textual data. The primary objective of sentiment analysis is to ascertain the prevailing polarity of the sentiments conveyed in the text, which may be classified as positive, negative, or neutral [42].

Several studies have utilized Twitter data for studying psychological health. Zhang et al. [38] developed a pipeline to monitor the trends of depressive users and analyzed depression levels. Fatimah et al. [43] used tweets posted by Tweeters from Indonesia to detect anxiety and other psychological issues. Some works have focused on the detection of specific psychological illnesses from posted tweets such as depression detection [44,45] and detection of post-traumatic stress disorder [46]. Roy et al. [47] investigated the effects of the cannabis drug on psychological health.

Regarding research on mental health using Arabic Twitter data, we have found only a limited number of studies. Alabdulkreem [48] proposed a deep-learning technique to predict depressive and non-depressive Arabic tweets in Saudi Arabia. Almouzini et al. [49] proposed a supervised predictive model to detect depression among Twitter posts in the Gulf region using sentiment analysis.

### Research Gap

Our work differs from previous research studies from a variety of perspectives including its particular focus, the nature of the dataset (data size, language, time period, and geography), the software design (the pipeline and approach for machine learning), the innovative methodology of using AI for discovering parameters, and the innovative methodology and design of finding associations between parameters and drugs.

## 3. Methodology and Design

In this section, our methodology and the design of our tool are explained. Figure 1 depicts the proposed system architecture. The architecture consists of five modules: data collection and storage, data preprocessing, parameter discovery, validation, and reporting and visualization. These modules will be covered in the subsequent sections. The methodology overview of the proposed tool will be discussed in Section 3.1. The architecture’s modules will be discussed in Section 3.2, Section 3.3, Section 3.4, Section 3.5 and Section 3.6.

Note that we have used contextual translations and made some adjustments to the translations of original Arabic tweets in order to make them more understandable to English readers. This may include changes to the order of the information in the tweet, the removal of unnecessary or redundant information, and the provision of summaries for tweets that are too long or contain unnecessary information. We have also sometimes omitted parts of the original tweets in order to protect the privacy of the tweeters. Note that Arabic tweets (typically true for any language) tend to be written in an informal style, so a literal translation may not always be clear or convey the intended meaning. Note also that in some tables in the paper, some search terms, or key terms detected by our machine learning models, may appear multiple times. This is because the original terms in Arabic may be different, but their English translations may be identical. The Arabic content (key terms, tweets) is not presented in this paper due to the publisher’s requirements. It can be found in an extended version of this paper made available on a preprint server [16].

### 3.1. Methodology Overview

This study proposes a big data and machine learning-based approach for the automatic discovery of parameters related to psychological health from Twitter data. The proposed approach focuses on psychological disorders in Saudi Arabia but can be applied to other diseases and languages. There are five components in the proposed approach: data collection and storage, data preprocessing, drugs for psychological health parameters discovery, validation, and visualization and reporting. The first step was to use a Python script with a specified search query and a set of keywords and Twitter hashtags related to psychological health in Saudi Arabia. A discovery module was then constructed for data analysis and detection of parameters using Latent Dirichlet Allocation (LDA) and the scikit-learn library. We discovered the parameters from three different perspectives (Drugs and Treatments, Causes and Effects, and Drug Abuse). The results for each perspective are discussed in detail in Section 4, Section 5 and Section 6. The discovered parameters are then presented visually through an intertopic distance map and keyword frequency diagrams. Finally, the results were validated internally and externally by tweets in our dataset and supporting scientific literature.

### 3.2. Data Collection

We collected Arabic tweets that are related to psychological health in Saudi Arabia using Twitter REST API and Tweepy. The data was obtained using various key terms and hashtags related to psychological health. For instance, the following key terms were used: depression, sadness, panic, mental illness, and others. Additionally, we used various hashtags such as depression month, social anxiety, social phobia, and others. A sample of the keywords used for data collection is as follows: suicide, social phobia, depression, depressed, sadness, fear, anxiety, obsessive, incantation, envy, panic, neurology, psychotherapy, mental health, psychological counseling, and mental illness. Some of the hashtags used include World Suicide Prevention Day, Suicide Awareness Month, suicide prevention, social anxiety, social phobia, depression, depression month, and seasonal depression. The list of Arabic key terms used in data collection can be found in an extended preprint version of this paper [16]. The data was collected from the 1 to the 31 of October 2022. Approximately, 1,048,575 tweets have been obtained. Tweets were retrieved from Twitter as JSON (JavaScript Object Notation) objects. Every tweet involved several attributes such as “full_text”, “created_at”, “id”, “place”, and “geo”. After that, we extracted these attributes and saved the result in an XLSX file. Duplicate tweets were removed based on Tweet “Id”.

### 3.3. Data Preprocessing

Data analytics requires the preparation of data as a critical ingredient. Data preprocessing involves a number of methods for cleaning, eliminating noise, improving quality, and, eventually, increasing accuracy. One of the libraries available for preparing textually based data is Natural Language Toolkit (NLTK). Preprocessing includes a number of steps including tokenization, normalization (replacing letters), stop word removal, and the elimination of irrelevant words and characters. Our first step in the preprocessing was eliminating all irrelevant characters and words such as numbers, URLs, different symbols (e.g., &, @, and #), English alphabets, emojis, etc. Moreover, we eliminated non-Arabic characters, repeating characters, and all various forms of punctuation symbols such as brackets and mathematical notations. The next step was tokenization and normalization in which we removed all different types of Arabic diacritics including single marks such as Fatha, Dammah, Kasra, Tashdid, and Sukun as well as double marks such as Tanwin Damm, Tanwin Kasr, and Tanwin Fath. Moreover, we used the normalizer to convert all different shapes of Alif, Yaa, and Taa Murbutah to the basic form bare Alif, dotless Yaa, and Haa, respectively. After that, we removed the list of stop words provided by the NLTK library with an additional list of words in dialectical Arabic developed by us; further details of data-preprocessing of Arabic tweets including a list of Arabic stop words can be found in our earlier work [35].

### 3.4. Parameters Discovery

In this section, we discuss the methodology for identifying psychological health parameters through topic modeling analysis of Twitter data. Modeling of topics is a frequently employed AI approach for data analysis and topic discovery, and it contains various algorithms that identify patterns and themes in a collection of documents by clustering word terms [48]. For topic modeling, one popular unsupervised learning approach is the LDA algorithm. It is a statistical technique for determining the topics that come up most frequently in a group of documents. It works on mapping a group of documents (such as tweets) into a group of themes or clusters, assigning each document a certain likelihood of being related to a specific topic. The parameter discovery was implemented on Google Colab platforms using various Python packages such as Scikit-Learn, Numpy, and Pandas.

We modeled the data from three perspectives: Drugs and Treatments, Causes and Effects, and Drug Abuse. We used a list of keywords to create a subset of the dataset and discover the parameters for each perspective. For instance, for the Drugs and Treatments perspective, we used names of antidepressants, painkillers, and medicines (e.g., Panadol). For the Causes and Effects perspective, we used multiple keywords such as side, effects, and cause. For the Drug Abuse perspective, we used multiple keywords such as abuse and extra. Most of the keywords are in Arabic and few in English because some tweets use some terms in English such as medicine names. We modeled each perspective into different clusters. After extracting the clusters, we allocated each tweet to its cluster based on the highest probability of the tweet association with a cluster. After that, we performed an analysis of the tweets and keywords in each cluster in which we looked at the keywords and examined the context of the keywords in each parameter. This enabled us to name each cluster based on the keywords and tweets using our domain knowledge. We iteratively refined clusters’ names using our domain knowledge and other quantitative measures. The process enabled us to eliminate irrelevant clusters and combine clusters that were similar. We eventually aggregated the parameters based on their common themes into macro-parameters that represent broader areas. This is done separately for each perspective.

### 3.5. Validation

The discovered parameters were validated internally and externally. For external validation of the data and parameters extracted from the Twitter data, we utilized academic papers, news articles, and online reports. To assess the validity of the discovered data and parameters, internal validation was carried out utilizing tweets from the gathered dataset.

### 3.6. Visualization and Reporting

In this study, we provide a variety of visualization methods of the parameters we have discovered. These are intertopic distance maps, taxonomies, and keyword frequency diagrams (both cluster-specific and corpus-wide). Python pyLDAvis package was used to compute and depict the terms frequency diagrams and distance maps [50,51]. The intertopic scaling and distances were computed utilizing the Jensen–Shannon divergence. The width of the bars in the diagrams of keyword frequency represents the frequency distributions at the topical and corpus levels, respectively. Matplotlib was one of the other Python libraries we used.

## 4. Results: Parameter Discovery for Psychological Heath (Drugs and Treatments)

This section focuses on the parameters discovered for the Drugs and Treatments perspective. Section 4.1 presents an overview of parameters and macro-parameters. Section 4.2, Section 4.3, Section 4.4, Section 4.5 and Section 4.6 explain the parameters in detail. The associations between the detected parameters and drugs are provided in Section 4.7.

As noted in the Introduction Section, we have translated the Arabic content (words and tweets) contextually and made adjustments to the original text, including changes to the information order and the removal of unnecessary or redundant information. We have also omitted parts of the original text that were not useful.

### 4.1. Overview and Taxonomy

We used a list of Arabic and English keywords to create a subset of the dataset and discover the parameters for the Drugs and Treatments perspective. The subset contains 6717 tweets. The LDA algorithm detected 30 clusters from the subset of the dataset. We merged similar clusters, discovered parameters, and categorized them into five macro-parameters.

The keywords used (translated into English) are as follows: medicine, drugs, pharmaceutical, medicinal, prescribe, prescription, dose, antidepressant, as anti (depression), anti (depressants), tranquilizer, milligrams, pill, pills, reliever, melatonin, Panadol, Rufenac, Celebrex, Ibuprofen, Acetaminophen, Brintellix, Duloxetine, Faverin, Seroxat, Lyrica, Remeron, Cipralex, Xanax, Benzodiazepine, Valium, Escitalopram, Leponex, Paroxetine, Bupropion, Imipramine, Haloperidol, Reserpine, Tetrabenazine, Clonazepam, Lorazepam, Diazepam, Amitriptyline, Nortriptyline, Mirzagen, Prozac, Serotonin, Cyproheptadine, Salipax, Tramadol, Wellbutrin, Letrozole, Cabergoline, Tranylcypromine, Gomood, Rhodiola, Ashwagandha, Duspatalin, and Omeprazole.

Table 1 provides a list of the detected parameters for the Drugs and Treatment perspective. Column 1 lists the macro-parameters. A total of five macro-parameters are present. The second column presents twenty-four parameters. Some of the parameters that are related to one another are merged. The cluster numbers (created by LDA clustering) are provided in the third column. In Column 4, the keywords’ percentage of the parameters are presented. The top 20 keywords related to each parameter are listed in the fifth column.

A taxonomy (see Figure 2) illustrating the Drugs and Treatments perspective was created using the parameters detected by our software. The parameters and their macro-parameters are displayed in the taxonomy. The macro-parameters Diseases and Disorders, Individual Factors, Social and Economic Factors, Treatment Options, and Treatment Limitations are represented at the first level. Second-level branches display the discovered parameters such as anxiety, sadness, poor concentration, etc.

Figure 3 presents the intertopic distance map and the overall term frequency of the top 30 keywords for the dataset of the Drugs and Treatments perspective. An intertopic distance map is a graphical representation of the relationships between topics (clusters) in a text or corpus. It is a useful representation which shows the overall structure of topics (clusters), their sizes, and how they are related to each other. The intertopic distances and the scaling for the set of intertopic distances are computed using the default options Jensen–Shannon divergence and principal components, respectively. The left-hand side of the figure depicts a representation of the 30 detected clusters, encompassing their sizes and interrelationships. The lower-left side of the diagram shows a key that denotes the size of the clusters. The blue bars on the right side give the top keywords’ overall term frequency. The highest frequency is for depression keyword, which is more than 3000. 

### 4.2. Diseases and Disorders

In this section, we discuss the parameters related to the macro-parameter Diseases and Disorders. Figure 4 shows the top 10 key terms according to term frequency (for further details see Section 3.6).

#### Postpartum Depression

This parameter is about postpartum depression which is a form of depression that develops in women after giving birth to a child. The parameter is represented by keywords such as depression, birth, gloom, death, mother, afflict, women, sadness, husband, hate, and postpartum. Several tweets in this parameter discuss the symptoms of this disease such as exhaustion and lack of energy, sleep disturbance, anorexia disorder, weakness in concentration, and thinking about death.

### 4.3. Individual Factors

In this section, we discuss the parameters related to the macro-parameter Individual Factors including anxiety, sadness, poor concentration, poor memory, loss of appetite, and fear of medicine. Figure 5 shows the top 10 key terms for each parameter in Individual Factors macro-parameter.

#### 4.3.1. Anxiety

This parameter relates to anxiety, a common emotion characterized by worry, nervousness, and unease about an uncertain outcome. It is a normal reaction to stress that everyone experiences at some point, but anxiety disorders are more than short-term worries or fears. Anxiety can have a negative impact on daily activities such as work, school, and interpersonal relationships [52]. Common indications of anxiety may include feeling uneasy, agitated, or restless; experiencing a sense of panic or dread; having an accelerated heart rate; hyperventilating; feeling fatigued; difficulty focusing on anything other than the current concern; and having difficulty sleeping [53]. There are various types of anxiety disorders, including generalized anxiety disorder, panic disorder, social anxiety disorder, and phobias [52]. This parameter covers various issues, including medicines; treatment plans and doses; and effects on sleep, pain, and overall health.

#### 4.3.2. Sadness

This parameter relates to sadness. It includes the following keywords: depression, treatment, sadness, time, psychological, anti (depression), symptoms, psychological, pills, treatment, deep, disappointed, hopes, wound, and heal. Many tweets in this parameter initiate sad thoughts. Some of the tweets are poems. We found several tweets that contain poems focusing on sadness due to love. They initiate sad thoughts in people; although people may enjoy it first, it can lead to severe depression and suicide like any other intoxication. It is well known that sad songs may give enjoyment to lovers, but they may also become a source of depression.

#### 4.3.3. Poor Concentration

The poor concentration parameter regards the difficulties in concentration and the issues related to it. This parameter contains the following keywords: depression, medication, treatment, disorder, anti (depression), self, causes, pills, diabetes, deficiency, anxiety, prescription, treatment, psychological, and dangerous. Although these keywords do not directly mention concentration, they are about diseases related to concentration. Most of the tweets are about causes of poor concentration including depression and anxiety.

#### 4.3.4. Poor Memory

The poor memory parameter discusses the negative effects of depression on memory and focus. People and experts discussed how depression affects memory and one’s ability to concentrate and remember. Moreover, some tweets have highlighted that some people fear using antidepressants because they think that it will cause issues with memory and concentration.

#### 4.3.5. Loss of Appetite

This parameter is about loss of appetite, which can happen because of depression. Some of the keywords in this parameter are biscuits, psychological, treatment, medication, depression, eating, taking, alone, light, coffee, chocolate, and food. Some people have mentioned loss of appetite among the negative effects of antidepressants.

#### 4.3.6. Fear of Medicine

This parameter regards fear of medicine. Individuals with this fear may feel intense fear and worry when considering taking medication or going to a doctor or hospital. The parameter covers some factors of fear, including the length of medicine (duration of treatment), side effects, fear about taking the incorrect medication, worry about being incorrectly diagnosed, concern about becoming dependent on medication, and fear about medical procedures or treatments.

### 4.4. Social and Economic Factors

Here, we cover the parameters related to the macro-parameter Social and Economic Factors including poverty, unemployment and insufficient finances, high cost of healthcare, loss of loved ones, forensic psychiatry, and social depression. Figure 6 shows the top ten key terms in each parameter.

#### 4.4.1. Poverty

This parameter relates to poverty as an economic factor that can cause mental health issues. This parameter captures various dimensions such as low household income, medical care, children, stealing, sadness, fear, depression, and psychological diseases. Some tweets under this parameter have highlighted how stealing money from poor people can have a devastating impact on their mental health. It can have a disastrous effect on the livelihoods of those affected, depriving them of the basic necessities such as food, shelter, and medical care that their families and children require for survival. Eventually, it can lead to feelings of despair, depression, and mistrust in others.

#### 4.4.2. Unemployment and Insufficient Finances

This parameter discusses inadequate finances and unemployment as social and economic factors for depression and mental health issues. This parameter includes various dimensions detected by our model such as work conditions, unemployment status, children, prison, depression, and committing suicide.

#### 4.4.3. High Cost of Healthcare

This parameter relates to the high cost of healthcare as one of the socioeconomic causes of depression. This parameter involves various dimensions such as low income; healthcare expenses; chronic diseases; healthcare expenses for elderly parents; and stress and pressure. Some tweets discussed how low-income elderly people with chronic illnesses, such as diabetes or hypertension, tend to incur greater expenses for medical services. This may result in inadequate housing and nutrition, which may worsen health outcomes. All of these factors may lead to higher healthcare costs for those who are poor. This can contribute to feelings of stress and pressure.

#### 4.4.4. Loss of Loved Ones

This parameter highlights one of the social causes of depression, the loss of loved ones. The parameter is represented by keywords such as pills, depression, period, feeling, lost, most important, depression, best, sleep, matter, medicine, even, life, living, death, friend, desire, and Iniesta. Some tweets related to the depression experience of the football player Iniesta who got depressed from the death of his close friend. The following tweet is an example: “When I was fighting depression, my best time was when I swallowed pills and went to sleep. Even hugging my wife was like hugging a pillow, without feeling.”.

#### 4.4.5. Forensic Psychiatry

This parameter is about forensic psychiatry, defined as “the branch of psychiatry that deals with issues arising in the interface between psychiatry and the law, and with the flow of mentally disordered offenders along a continuum of social systems” [54]. It is “applied to legal issues in legal contexts embracing civil, criminal, correctional or legislative matters” [55]. The parameters involve treatment services in legal proceedings, the role of forensic psychiatry, consultation services, and the efficiency of legal proceedings.

#### 4.4.6. Social Depression

This parameter is about social depression. This parameter highlights various dimensions such as stressful life events, stigmatization of medicines, weight increase, and lifestyle. This parameter emphasizes the fact that society is living in a time when the cost of living and healthcare has increased, and high achievement has become a necessity leading to social depression and anxiety.

### 4.5. Treatment Options

The parameters associated with Treatment Options macro-parameter are discussed in this section. Figure 7 depicts the top 10 key terms based on term frequency.

#### 4.5.1. Walking

This parameter discusses walking as a treatment for psychological diseases. It is represented by keywords such as prescribe, body, walking, negativity, psychological, energy, nature, anxiety, needs, medications, diseases, fear, equivalent, work, painkillers, emptying, endorphins, sedatives, and reduce. The tweets in this parameter discuss a range of benefits of walking such as triggering the body’s whole muscular system and reducing relapses of mental illnesses.

#### 4.5.2. Optimism

This parameter is regarding optimism, which is a psychological approach or outlook that emphasizes the beneficial aspects of life and anticipates positive results. Optimism is often regarded as a form of resilience, allowing individuals to manage challenging circumstances and recover from adversity [56]. This parameter involves optimistic poems, happiness, fighting sadness, and environmental influences.

#### 4.5.3. Good Company

This parameter is about good company and the importance of good friends for mental health issues. Some of the keywords for the parameter are depression, anti (depression), best, friend, anti (depressants), normal, good, and defect. People discussed how good friend can be as an antidepressant for depression. There are many tweets in this parameter such as the following, “Best antidepressant: (a good) Friend”.

#### 4.5.4. Pendulum Technique

This parameter focuses on the pendulum technique. It contains the following keywords: fear, then, question, pendulum, yourself, effectiveness, know, answer, write, ask, feelings, attachment, ready, mention, answer, sharp, intention, depression, anti (depressants), and sun. It was detected as a treatment for psychological issues.

#### 4.5.5. Spirituality

This parameter covers spirituality as a treatment for sadness and depression. People discussed how spirituality is used to treat people from sadness and depression. For example, doing good actions, remembering God, and praying the morning prayer.

#### 4.5.6. Antioxidants

The antioxidants parameter focuses on the role of antioxidants in fighting depression and mental illnesses. It is represented by key terms such as coffee, psychological, depression, oxidation, treatment, anti (depression), condition, people, helps, most, moods, relieve, improve, anti (oxidants), richness, fruits, combined, plus, and vegetables. The tweets in this parameter have discussed natural sources of antioxidants. Moreover, many tweets have mentioned how coffee is rich in antioxidants and how it can help relieve depression and improve mood state.

#### 4.5.7. Painkillers and Antidepressants

The painkillers and antidepressants parameter highlights the difference between painkillers and antidepressants in terms of their use. This parameter contains the following keywords: depression, medications, disease, treatment, patient, psychiatric, medication, psychological, anti (depression), instead of, doctor, depression, for a patient, Cipralex, painkiller, body, give, Celebrex, and hurt. Some tweets have mentioned that antidepressants can be prescribed for physical illnesses. It is not clear from the tweets why an antidepressant is prescribed. Doctors may see some symptoms of depression.

#### 4.5.8. Community-Supported Therapies

This parameter is about community-supported therapies. This parameter includes various dimensions such as community support groups, society programs, faith, successful treatment factors, etc. Here is an example of a related tweet: “Psychological diseases involve a set of genetic, familial and social factors, and therefore recovery from them also requires a combination of all these factors, such as regular use of medication, adherence to healthy habits and lifestyle “sports”, family support, community awareness and embrace and not to reject those who suffer from it, or to stigmatize them as weak or lacking in faith!”.

#### 4.5.9. Psychotherapy and Medication

This parameter is about the types of medical treatments for psychological illnesses, which are Cogitative Behavioral Therapy (CBT) and the use of medications/drugs. A number of tweets discuss psychotherapy and medication. For example, a tweet stated that “in mental illness, each case is different from the other. And depends on the condition and depression degrees. Medicines are used in severe cases and behavioral therapy benefits most people. Therefore, first, you must visit a doctor, who will examine you and let you know whether you need to take medicine or undergo behavioral therapy”. In addition, some tweets highlighted the importance of lifestyle for mental health.

### 4.6. Treatment Limitations

The parameters related to the macro-parameter Treatment Limitations are covered in this section. Figure 8 shows the top 10 key terms, in each parameter, based on term frequency.

#### 4.6.1. Antidepressant Limitations

This parameter discusses antidepressant limitations. It is represented by keywords including depression, medicine, truth, relieve, reality, natural, dealing, mind, crises, right, exaggerating, delight, emotion, happiness, nervousness, etc.

#### 4.6.2. Negative Effects of Antidepressant

The parameter is about the negative effects of antidepressants. It is represented by keywords such as depression, medicine, anti (depressants), people, psychological, sadness, possible, pill, condition, psychological, actually, disease, nervousness, causes, etc. There are many tweets that mention the side effects of antidepressants. As previously noted, we have translated the Arabic content (words and tweets) contextually and made adjustments to the original text, including changes to the information order and the removal of unnecessary or redundant information. We have also omitted parts of the original text that were not useful. Here is an example tweet: “If the psychiatrist is incompetent, he will give the patient pills that ruin a person’s life”. A tweeter stated that one of her siblings committed suicide after a doctor convinced him that his depression doesn’t have a solution and there is no treatment for it. In addition, in another case, it was reported in a tweet that someone’s close relative was prescribed so many strong pills that if he forgot to take them for a day or two he would have a bout of screaming and crying.

A number of tweets mentioned some diseases that were detected by our tool as side effects of antidepressants such as obesity, drowsiness, bruxism, and attention-deficit/hyperactivity disorder (ADHD). Some tweets have discussed the positive sides of using antidepressants. For example, the following tweet: “… some mental illnesses are chronic like some physical illnesses such as diabetes and high blood pressure. Therefore, you may need to take antidepressants for long periods or all of your life”. Several tweets have mentioned other ways of treating depression such as electroconvulsive therapy (ECT), St. John’s wort, and magnesium.

### 4.7. Parameter-Drug Associations (Drugs and Treatments)

In this section, we provide the associations between the detected parameters and drugs for the Drugs and Treatments perspective. These are shown in Figure 9. For example, for the sadness parameter the associated drugs included Prozac, Cipralex, Remeron, and Bupropion. These are antidepressants and their association with the sadness parameter shows that sadness, which is related to depression, may have led to the use of these drugs. Sadness can be a symptom of depression or a cause for it. Gartlehner et al. [57] discussed how second-generation antidepressants could be used to treat Seasonal Affective Disorder (SAD) among adults and reduce the severity of its symptoms, including sadness. Leventhal [58] discussed how sadness can cause depression and mentioned that sadness is a normal emotion that can result in depression if it is not managed appropriately. The parameters and macro-parameters are the same as those listed in Table 1 and discussed earlier in this section (Section 4). Similar to Figure 2, in Figure 9, the first-level branches show the macro-parameters and the second-level branches show the detected parameters. The drugs associated with each parameter are shown on the third-level branches, where available.

These parameter-drug associations can be discovered automatically as follows: we built a vocabulary of all medicines used for the treatment of psychological illnesses, and we searched for these medicines against tweets in each parameter and recorded the associations with the drugs found through the search for each parameter.

## 5. Results: Parameter Discovery for Psychological Heath (Causes and Effects)

This section discusses the parameters discovered for the Causes and Effects perspective. An overview of parameters and macro-parameters is provided in Section 5.1. The parameters are explained in Section 5.5, Section 5.3, Section 5.4 and Section 5.5. Section 5.6 presents the associations between the detected parameters and drugs.

### 5.1. Overview and Taxonomy

We created a list of Arabic keywords to build a subset of the dataset and identify the parameters for the Causes and Effects perspective. The dataset obtained contains 88,566 tweets. The parameters and other information on the Causes and Effects perspective are provided in Table 2. The taxonomy is given in Figure 10.

The keywords employed are side, effects, because of, cause it, it causes, caused by, cause, brought, result, weight, my weight, cholesterol, disorders, lethargy, migraine, appetite, metabolism, memory, concentration, dizziness, sleep, insomnia, headache, crying, stomach, hyperactivity, attention, deficit, depression, and addiction.

### 5.2. Diseases and Disorders

In this section, the parameters that belong to the macro-parameter Diseases and Disorders are discussed including attachment disorder, insomnia, and obsessive-compulsive disorder (OCD). Figure 11 depicts the top 10 key terms, in each parameter, according to term frequency (for further details see Section 3.6).

#### 5.2.1. Attachment Disorder

This parameter is about attachment disorder, which is a form of mental illness or behavioral condition that interferes with a person’s capacity to establish and sustain relationships. It relates to the challenges involved in understanding emotions, expressing affection, and placing one’s trust in others. The parameter is represented by keywords such as psychological, health, family, live, your life, hospital, person, reality, well-being, success, locked up, lost, attachment, money, etc. People discussed that someone should avoid excessive attachment to loved ones as it can destroy person’s life. Furthermore, a case of a celebrity who was deceived by a loved one was discussed.

#### 5.2.2. Insomnia

This parameter focuses on insomnia, which can be a cause or an effect of other psychological issues. This parameter is characterized by keywords such as sleep, sadness, anxiety, doctor, eye, fear, depression, symptoms, fear, diaspora, matter, etc. People discussed different reasons for insomnia such as excessive worry, fear, depression, anxiety about events or people, sadness, excessive thinking, exhaustion, or loss and nostalgia.

#### 5.2.3. Obsessive-Compulsive Disorder (OCD)

This parameter regards obsessive-compulsive disorder (OCD), which is a prevalent mental health problem characterized by compulsive behaviors and obsessive thoughts. People discussed the symptoms of OCD, the causes and treatment. For example, someone tweeted: “Obsessive-compulsive disorder is the control of an idea that its owner knows is absurd, forcing him to repeat actions, such as making sure the door is locked, cleanliness, or purity, to a degree that may affect the productivity of the individual. This indicates underlying anxiety and can be treated with some medication and dialogue...”.

#### 5.2.4. Post-Surgery Depression

This cluster is about post-surgery depression, and it focuses on surgeries as a cause of depression. The tweets associated with this parameter are mostly related to the depression that occurs after Sleeve gastrectomy surgery because the stomach is restricted to a certain food, and this has a negative effect, such as feeling lonely or that the person cannot go out and eat a variety of foods like before. However, depression can happen as a side effect of any other surgery.

#### 5.2.5. Chronic Physiological Diseases

The chronic physiological diseases parameter discusses various diseases that could lead to depression. The following keywords were detected by our model: depression, depression, cause, psychological, sick, chronic, king, medical, brain, fear, diseases, Salman, suffering, surgical, cause, city, relationship, psychological, nerves, and compensate. When a person suffers from a disease that affects his ability to move and could lead to some changes in lifestyle, this could result in depression. A tweet mentioned five chronic diseases which cause depression and sadness including diabetes mellitus, arthritis, heart disease, kidney failure, and thyroid gland. Some other tweets linked COVID-19 infection to a range of chronic neuropsychiatric disorders, including depression, memory problems, and Parkinson’s disease-like disorders.

### 5.3. Individual Factors

We highlight here the parameters under the macro-parameter Individual Factors. There are eight parameters. Figure 12 depicts the top 10 keywords, in each parameter, based on term frequency.

#### 5.3.1. Fear

This parameter is about fear as a cause or effect of psychological illnesses. Our model detected the following keywords: leave, care, fear, weight, gain, about you, subject, sleep, diseases, poverty, keep away, think, difference, fear, doctor, health, face, your fear, and sources. The tweets highlighted different kinds of fear including fear of losing persons, fear of diseases, fear of poverty, and others.

#### 5.3.2. Sadness

This parameter is about sadness which could be a symptom, a cause, or an effect of psychological diseases. It is represented by keywords including world, depression, wish, real, people, complete, me, normal, age, try, need, needs, work, fear, person, years, time, stay, etc. This parameter is similar to a parameter covered in the previous perspective. For more details see Section 4.3.2.

#### 5.3.3. Loneliness

This parameter is about loneliness, which is characterized by keywords such as wish, heart, alone, sadness, complete, pass, loneliness, mind, stage, fear, focus, nights, human, thinking, anxiety, unknown, details, compensate, trust, and calm down. Someone tweeted: “I hope that God will compensate me for all the nights of loneliness, sadness and misery, and reassure my heart …”.

#### 5.3.4. Lacking Passion

This parameter is about people who lost their sense of value and pleasure in everything and wish for death. This parameter includes the following keywords: depression, want, need, myself, times, moment, desire, overwhelming, disappear, the world, have, presence, heavy, exist, feel, want, depression, sadness, and view. People discussed different symptoms associated with lacking passion such as a feeling of helplessness, low energy and exhaustion, constant pain, and the feeling of guilt. Other tweets have mentioned other symptoms including a lack of self-esteem, self-loathing, lack of focus, loss of hope, and the desire to disappear.

#### 5.3.5. Suppressing Emotions

This parameter is about the suppression of emotions either positive or negative, which can lead to depression and other psychological illnesses. Based on our model, the following keywords were detected: sadness, sorrow, physical, cause, after, able, personality, disease, experience, sleep, possible, upset/angry, need, your chest, was not, wish, tell, say, inside, and live. Some tweets highlighted some of the effects of suppressed emotions such as anxiety, depression, and other stress-related illnesses. Other tweets mentioned the importance of discussion and expressing emotions for psychological health.

#### 5.3.6. Negative Emotions

The negative emotions parameter is about people who talk about and share their personal negative experiences and generalize them, causing depression for themselves and others in society. It is represented by keywords such as depression, condition, people, friendliness, because, human, life, depression, psychological, crying, sleep, conversation, life, yourself, have, sadness, anxiety, permanent, phrase, and love.

#### 5.3.7. Devil (Negative Thoughts)

This parameter regards the devil and negative thoughts. It is characterized by keywords such as most important, sadness, anxiety, whirlpool, fear, heart, devil, life, comfortable, bad, sorrows, stable, caused, current, last, past, make, tense, and destroy. People discussed how the devil negatively affects people’s mental health. The following tweet is an example: “Remember that one of devil’s most important goals is to cultivate sadness and fear in the heart, so that he does not make you stable or comfortable, but rather discontented, anxious, and pessimistic. He links you to the past, its pains, and the sorrows it causes you, and links you to the future, its fears and anxieties; To make you always in a tense spiral and mistrust, and his goal is to destroy your current moment and spoil your life.”.

#### 5.3.8. Lacking Inner Peace

This parameter is about lacking inner peace. The following keywords were detected by our model: life, peace, anxiety, insomnia, stay away, in you, people, many, things, topic, anger, inside me, focus, your Lord, struggle, urgency, fear, anxiety, psychological, and joy. This parameter focuses on the importance of inner peace for fighting depression. People discussed different things such as how to get inner peace by avoiding passing judgment on people. Following is an example tweet: “If you do not feel peace within you, you will find many things in life that cause you anger, chaos, grumbling, anxiety, and conflict. How do I find peace inside me? get closer to your Lord; avoid passing judgment on people; stay away from focusing on any disturbing topic; live life with grace, not with complexity.”.

### 5.4. Social and Economic Factors

There are five parameters under the Social and Economic Factors macro-parameter. Figure 13 shows the ten top 10 keywords, in each parameter, based on term frequency.

#### 5.4.1. Study

This parameter covers various study-related issues which could cause psychological illnesses such as studying for long hours, studies-related depression, and bullying in schools. The parameter contains the following keywords: concern, problem, subject, permission, fear, cause, psychological, lead, schools, academic, level, impact, delay, space, going, coming, elite, to school, disability, and counsellors. The tweets discussed the causes of psychological illness. For instance, the following tweet highlights different causes of psychological illnesses and some solutions which don’t lie in drugs: “When the psychologist’s tweets highlight how some psychological disorders, such as depression, anxiety, etc., develop as a result of people’s exposure to psychological trauma, abused childhood, or some social problems such as divorce and others. It is natural to find that the solution to these problems does not lie in drug treatment”.

Furthermore, a tweet stated a list of disorders which are related to certain causes. These disorders include anxiety disorders, especially panic attacks and anxiety about disease, depression and mood disorders, traumatic disorders, personality disorders, dissociative disorders, and internal psychological struggle due to social pressure. Another tweet highlighted various socioeconomic causes of depression and psychiatric disturbances. A tweet reported that “the poor economic state of the family may cause social problems and bad psychological effects that lead to excessive thinking and eventually lead to mental illnesses”.

A number of tweets reported that universities and schools cause fear and depression. Moreover, several tweets discussed the issues of bullying in schools and how it affects the academic progress of students. For example, a tweet mentioned that bullying in schools can cause depression, anxiety, social shyness, social phobia, and eventually delay in the academic level.

#### 5.4.2. Work

This parameter focuses on work as a cause of psychological issues. Among the keywords that our model detected are depression, limit, need, possible, permanence, depressed, length, fear, offender, no one, praise be to God, literally, still, life, sufficiency, society, coming, and deficiency. People discussed how long working hours affect mental health and how leaving very little time for family and social relationships can result in depression and family breakup.

#### 5.4.3. Lifestyles

This parameter is about the lifestyle as a cause of psychological illness including eating and thinking patterns. The parameter is represented by keywords such as time, depression, sadness, cause, grace, speech, problems, know, silence, understood, inside, pretended, stupid, committed, smiled, answered, wellness, weight, in relation to, and hospital. Here are some example tweets about maintaining a good lifestyle. For example, the following tweet: “Most people write about pain and talk about fatigue until their minds are programmed to be depressed and think negatively which cause them illnesses”.

Following is another example tweet: “Malnutrition is the cause of mental illness, which can be treated with diet, exercise, cupping, and good company rather than medicine. The consumption of indomie, soft drinks, and drinks containing stimulants causes fear. Alcohol, smoking, and sweets cause anxiety and depression.”.

#### 5.4.4. High Cost of Healthcare

This parameter is about the high cost of healthcare as a socioeconomic factor for psychological illnesses. This parameter includes the keywords depression, knew, make, have, session, psychological, period, good, for depression, seasons, diseases, suffering, fear, difficult, home, street, family, and life. This parameter is similar to a parameter covered in the previous dimension. For further details see Section 4.4.3.

#### 5.4.5. Seasonal Depression (Seasonal Effective Disorder)

This parameter is about seasonal depression which is a type of depression that occurs as a result of the change of seasons. This parameter is represented by keywords such as depression, Saturday, gloom, depression, severe, I have, birth, feel, winter, weather, spray, period, know, month, offender, people, cause, feel, atmosphere, and inside. In tweets and keywords, different types of depression were mentioned such as post-weekend depression, postpartum depression, and winter depression.

### 5.5. Treatment Options

Figure 14 displays the most frequent keywords in each parameter in the Treatment Options macro-parameter.

#### 5.5.1. Emotional Release (Psychotherapy)

This parameter regards emotional release (catharsis) as part of psychotherapy. The following keywords were detected by our model: depression, life, fear, hair, remove, cut, winter, sleep, name, family, wake up, satiate, inside, side, entered, smell, bring, come, answer, and people. Many tweets have talked about cutting hair as a way of emotional release. The following tweet, for example: “Cutting hair removes 100% of life’s depression”.

#### 5.5.2. Good Friends

This parameter is about good friends and is described by the following keywords: depression, better, anxiety, person, can, deeper, seriously, kidding, inside you, collect, spontaneity, quest, reach, continuity, wonderful, include you, the two things, the mother, cause, and not happened. People discussed the importance of good friends for psychological health. Following is an example tweet: “When you talk to good friend while you are in a state of anxiety and fear, you become reassured because of his deep words and great actions”.

#### 5.5.3. Spirituality

This parameter covers spirituality as a treatment for psychological illnesses. This parameter is similar to a parameter covered in the previous perspective. For further details see Section 4.5.5.

#### 5.5.4. Surgery

This parameter is about surgery as a treatment for psychological diseases. Among the keywords that our model detected are operation, depression, sadness, hours, surgery, success, suffering, future, sleep, psychological, medical, patient, mood, Salman, natural, first, thinking, anxiety, excess, and permanent. Several tweets have talked about the success of a surgical operation to treat a patient suffering from chronic depression.

### 5.6. Parameter-Drug Associations (Causes and Effects)

Similar to Section 4.7, here we provide the associations between the detected parameters and drugs for the Causes and Effects perspective. These are shown in Figure 15. For example, for the insomnia parameter, the associated drugs include Cipralex which is an antidepressant. Their association with the insomnia parameter shows a direct relationship between insomnia and depression in which either one of them can be a trigger for the other [59]. Insomnia, for example, may raise a person’s risk of developing depression tenfold compared to people who sleep well at night. On the other hand, depression is linked to sleep problems like getting less beneficial slow-wave sleep each night [60]. Moreover, the association between melatonin and the insomnia parameter is because people commonly use it for insomnia conditions [61].

Furthermore, we found that some painkillers, such as Panadol, are associated with the Insomnia parameter. The research highlights that the use of painkillers may be associated with an increased risk of developing insomnia. Opioid use is likely to be a contributing factor to insomnia due to the sedative effects of opioids, which can make it difficult to fall asleep and stay asleep [62]. Furthermore, long-term opioid use has been associated with changes in sleep patterns and disruption of the circadian rhythm, both of which can lead to insomnia [62].

It is possible that individuals who experience pain or headaches and have difficulty sleeping may benefit from the use of painkillers. Research has demonstrated that individuals suffering from chronic pain often experience sleep disturbances [63], including insomnia, which can intensify the pain [64].

## 6. Results: Parameter Discovery for Psychological Heath (Drug Abuse)

In this section, we discuss the parameters discovered for the Drug Abuse perspective. An overview of parameters and macro-parameters is provided in Section 6.1. The parameters are explained in Section 6.2. In Section 6.3, the associations between the detected parameters and drugs are provided.

### 6.1. Overview and Taxonomy

In this section, we discuss the Drug Abuse perspective. We employed a list of keywords to build a subset of the dataset and identify the parameters for the Drug Abuse perspective. The following is a translation of the list of keywords utilized: abuse, mood, trance, without a recipe, pill, and extra. The dataset that we got after filtering data contains 2,701 tweets.

The LDA algorithm detected 30 clusters for the Drug Abuse perspective. We excluded twenty clusters from the results as they were irrelevant to the focus of this perspective. We merged similar clusters, discovered parameters, and categorized them into five macro-parameters. Table 3 has structure similar to Table 1 and Table 2.

Using the discovered parameters for Drug Abuse perspective, a taxonomy was created (see Figure 16).

### 6.2. Drug Abuse

In this section, we discuss the parameters related to the macro-parameter Drug Abuse. Figure 17 shows the top 10 key terms, in each parameter in the Drug Abuse perspective, according to term frequency (for further details see Section 3.6).

#### 6.2.1. Bipolar Disorder

This parameter relates to bipolar disorder, known as manic depression, which is defined as “a mental illness that causes unusual shifts in a person’s mood, energy, activity levels, concentration, and ability to carry out day-to-day tasks” [65]. It involves various dimensions such as depression degree, mood swings, sadness waves, tears, and others.

#### 6.2.2. University Exams

This parameter regards university exams and is described by the following keywords: depression, condition, fine, pills, work, tried, unfortunately, help, peace, family, suicide, prison, answer, mercy, cut, tired, have, difficult, and see.

#### 6.2.3. Death of Loved Ones

This parameter relates to the death of loved ones as a cause of drug abuse. The following keywords were detected by our model: pills, depression, matter, feeling, period, depression, life, better, more important, medicine, even, death, lived, lost, sleep, go, desire, resistance, Kharkhi, etc. Some tweets relate to the depression of football player Iniesta and the death of his close friend which caused him depression.

#### 6.2.4. Addiction

This parameter discusses addiction, known as substance use disorder. It is a condition characterized by an impaired ability to control the use of legal or illicit drugs, alcohol, nicotine, or other substances, resulting in changes in brain function and behavior [66]. This parameter covers narcissism and addiction, health and risk factors, ecstasy, happiness, and others.

#### 6.2.5. Suicide

This parameter is about abusing drugs and committing suicide as an effect of psychological issues. The following keywords were detected by our model: psychological, potion, treatment, heart, one, long, love, doctor, take, pass, fear, depression, bad, etc. Some people mentioned that they have tried to commit suicide by using an overdose of medicine.

#### 6.2.6. Flakka Drug

This parameter is about the Flakka drug. The following keywords were detected by our model: depression, fear, love, potion, intensity, newness, intake, feeling, problem, desire, alone, therefore, withdrawal, dope, lethargy, drug, to withdraw, attempt, symptoms, and depression. Many tweets mentioned that this drug is spread among young people and the reason for its spread is that it is cheap. People have also discussed the effects of using the Flakka drug such as hallucinations, madness, strange behavior, loss of control over mental abilities, and a mad start to a dark path. Many tweets have also mentioned the withdrawal symptoms of the drug such as feeling lethargy and suffering severe depression.

### 6.3. Parameter-Drug Associations (Drug Abuse)

This section highlights the associations between the detected parameters and drugs for the Drug Abuse perspective. Figure 18 shows a taxonomy of associations between detected parameters and the drugs detected automatically by our tool. For example, in the figure, the Flakka drug is associated with the Flakka Drug parameter, which is a dangerous synthetic cathinone [67]. In addition, the melatonin drug is associated with the addiction parameter. This could be because melatonin can be used in addiction management [68]. Research has demonstrated that melatonin can reduce the pleasurable effects of drugs, decrease drug-seeking behavior, and decrease the relapse rate. Furthermore, melatonin may play a role in controlling stress responses related to drug addiction [69]. Escitalopram is also associated with the addiction parameter which is an antidepressant and it can be used in the recovery stage from addiction [70].

## 7. Discussion

In this research, we proposed a big data and machine learning-based approach for the automatic discovery of parameters related to psychological health from Twitter data. The parameters are discovered from three different perspectives: Drugs and Treatments, Causes and Effects, and Drug Abuse. Moreover, we automatically discovered associations between the parameters and drugs. The parameters were discussed in detail in Section 4, Section 5 and Section 6, respectively. We discussed the use of Twitter to automatically discover what drugs are used for psychological health, what the causes and effects of psychological issues are, what the side effects of drugs are, and how drugs are abused.

We discovered twenty-four parameters from the Drugs and Treatments perspective and grouped them into five macro-parameters: Diseases and Disorders, Individual Factors, Social and Economic Factors, Treatment Options, and Treatment Limitations. A total of twenty-two parameters were detected from the Causes and Effects perspective and we grouped them into four macro-parameters: Diseases and Disorders, Individual Factors, Social and Economic Factors, and Treatment Options. We detected six parameters from Drug Abuse perspective, namely, bipolar disorder, university exams, death of loved ones, addiction, suicide, and flakka drug.

A multi-perspective view of psychological health data is depicted in Figure 19. It is a combination of all three perspectives: Drugs and Treatments, Causes and Effects, and Drug Abuse. It includes six macro-parameters: Diseases and Disorders, Individual Factors, Social and Economic Factors, Treatment Options, Treatment Limitations, and Drug Abuse. We merged similar macro-parameters together. For example, we have two Diseases and Disorders macro-parameters, one from the Drugs and Treatments perspective with one parameter (postpartum depression), and another one from the Causes and Effects perspective with five parameters (attachment disorder, insomnia, obsessive-compulsive disorder (OCD), post-surgery depression, and chronic physiological diseases). We merged all these parameters in one Diseases and Disorders macro-parameter as shown in Figure 19.

This work makes important theoretical and practical contributions to the area. The earlier research (see Section 2) has looked into the relation between physical illnesses and mental health, specific mental health disorders and factors, effects of education on mental health, COVID-19 and mental health, machine learning in mental health, and the use of Twitter data in mental health. This study offers a comprehensive examination of mental health, including causes, treatments, and the impact of drug use and abuse, as seen on Twitter and discussed by both the public and health professionals. Additionally, this study identified associations between various drugs and mental health. This is the first study to take such a holistic approach to understand mental health.

This work sheds new light on the social and environmental factors that impact mental health and makes significant contributions to the field of toxics. Through the use of big data and machine learning, the study was able to identify the root causes of mental health issues from Twitter data and analyze parameters related to drugs and treatments, causes and effects, and drug abuse. As a result, this work provides a more comprehensive understanding of the complex social and environmental factors that contribute to mental health issues. This research seeks to identify and address the harmful social and environmental factors that can cause or worsen mental health issues, making it an important contribution to the field of toxics. Furthermore, it provides valuable insights into the relationship between mental health and various factors, which has the potential to inform the development of effective interventions that can improve public health and social sustainability. Our approach of focusing on different perspectives and aggregating identified parameters into macro-parameters is a powerful method that allows for a more nuanced and detailed understanding of mental health and its contributing factors. By identifying these factors, this study has the potential to contribute to the development of strategies to reduce exposure to harmful substances and mitigate the negative impacts of social, economic, and environmental factors on mental health. Therefore, the findings of this study have significant implications for the improvement of public health and social sustainability.

The findings have the potential to open new avenues for identifying drug use and abuse for mental health, as well as other micro and macro factors related to mental health through social media. The methodology can also be applied to other diseases and may have the potential for forensic toxicology research. However, more research is needed to fully explore the potential of social media for forensic purposes. Moreover, our approach can serve as an autonomous real-time surveillance system that captures crucial system parameters related to mental health, socio-economic factors, and the environment. By detecting opportunities, challenges, and risks, we can take proactive steps toward optimizing society for better, sustainable health outcomes. Moreover, our system’s ability to monitor and analyze a variety of factors in real-time allows for swift and effective action to be taken in response to potential threats to mental health. By continually tracking and assessing these parameters, we can proactively identify potential risks and take steps to mitigate them. This not only improves individual well-being but also helps to create a healthier, more resilient society.

The work presented in this paper is the beginning, many more works are needed to investigate the potential of social media for forensic and other purposes. This research is part of our broader work on data-driven parameter discovery from Twitter and other data sources applied previously to different research areas including the education sector in KSA during COVID-19 [35], the discovery of cancer-related healthcare services [71], families and smart homes [72], transportation [73], tourism [74], multi-generational labor markets [75,76], and COVID-19 governance measures [77].

This work is scientifically valid as it uses a systematic approach to gather data from Twitter and applies big data and machine learning techniques to analyze the data. The study collected a large sample of over one million tweets in Arabic about psychological health in Saudi Arabia. The use of a large sample increases the statistical power of the study, making it more representative of the population of interest. The study used machine learning algorithms to discover parameters related to mental health from the tweets, which ensures a consistent and replicable approach to the data analysis. The methodology used in the study is comprehensive and transparent. The paper provides a detailed account of how the study was conducted, including the data collection process, machine learning algorithms, and the parameters discovered. This allows other researchers to replicate the study and verify the results. Furthermore, this study identifies the potential for extending the methodology to other diseases and forensic toxicology, further demonstrating the robustness of the methodology. Our methodology and results are applicable to other research domains beyond mental health; therefore, our findings have a level of generality and transferability. This implies that results are not limited to the specific sample, population, or context used in this paper but can be extended to other contexts or populations. This study’s results provide a comprehensive account of mental health from different perspectives, including drugs and treatments, causes and effects, and drug abuse. This study identified 52 parameters from different perspectives related to mental health, which were further aggregated into six macro-parameters. This study’s findings were based on social media discussions by both the public and health professionals, which provides insight into the social and environmental factors that impact mental health. Furthermore, this study identified the associations between different drugs and mental health, which has implications for the effectiveness and risks of medication for treating mental health issues. In conclusion, this study’s comprehensive methodology, large sample size, and use of machine learning algorithms make it scientifically valid. This study’s results provide a valuable contribution to understanding mental health issues from a social and environmental perspective, and the potential for extending the methodology to other diseases and forensic toxicology provides a promising avenue for future research.

## 8. Conclusions

Mental health issues can have significant impacts on individuals and communities, and addressing root causes can help prevent mental health problems. The big data and machine learning approach proposed in this paper can be used to automatically discover parameters related to mental health from Twitter data, including information on drugs and treatments, causes and effects, and drug abuse. This can provide a comprehensive understanding of mental health as seen on social media, discussed by the public and health professionals, and can also identify associations with different drugs. The methodology can be extended to other diseases and has the potential for discovering evidence for forensic toxicology from social and digital media. Additional research is necessary to fully explore the potential of social media for forensic purposes; this paper is just the beginning and it will form our future work.

The conclusions drawn in this work are confirmed by scientific data and objectivity in several ways. First, the paper identifies mental health as a significant issue with impacts on individuals and communities and highlights the need for a holistic approach to understand its root causes. This understanding is supported by scientific research on mental health and the impacts of social and environmental factors on mental health outcomes. Second, we propose a data-driven approach for discovering parameters related to mental health from Twitter data. The use of big data and machine learning tools is a well-established methodology for data analysis and can provide insights into complex problems, including mental health. Third, we report the results of this study that collected over one million Arabic tweets about psychological health in Saudi Arabia. The use of Twitter data provides an objective and real-time view of mental health issues, as discussed by the public and health professionals. The analysis of the Twitter data provides a comprehensive account of mental health, causes, medicines and treatments, mental health and drug effects, and drug abuse.

Although our work on using big data and machine learning to automatically discover parameters related to mental health from Twitter data is innovative and insightful, there are limitations to the study that must be considered. Firstly, the study solely relies on Twitter data, which may not represent the entire population and may suffer from selection bias. Additionally, the study only focuses on Arabic tweets related to mental health in Saudi Arabia during a specific period, which may limit the generalizability of the findings. To address this limitation, more data should be collected and analyzed in Saudi Arabia and internationally. Secondly, the study collected a substantial number of tweets, but further research is needed to capture relevant tweets comprehensively, such as through additional keywords and hashtags. Thirdly, the research focused on three perspectives, and more research is needed to ensure that all perspectives are captured to provide more comprehensive information. Fourthly, there is a need to incorporate other sources of digital media and scientific data to enhance the diversity and richness of the discovered information. Finally, while machine learning algorithms are powerful tools for data analysis, they are not infallible and may produce inaccurate or biased results. We are confident that all the limitations we have identified can be overcome in our future work. As we move forward, we invite the community to join us in improving the proposed approach and enhancing its robustness and impact. Working together, we can achieve our goals and realize the full potential of our efforts.

## Figures and Tables

**Figure 1 toxics-11-00287-f001:**
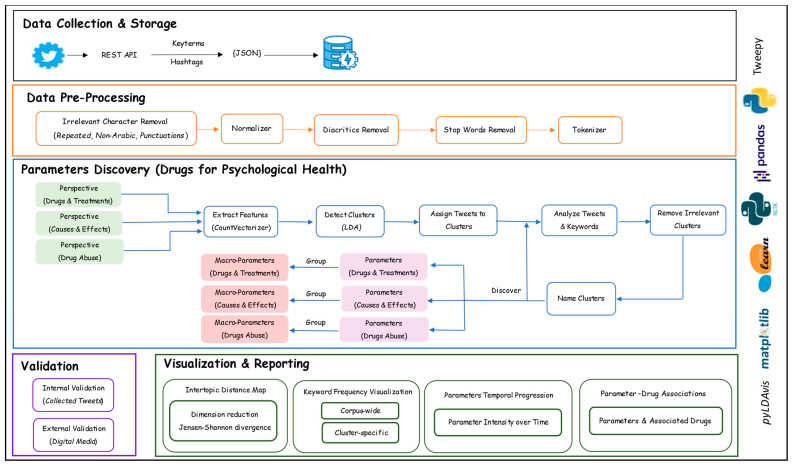
System Architecture.

**Figure 2 toxics-11-00287-f002:**
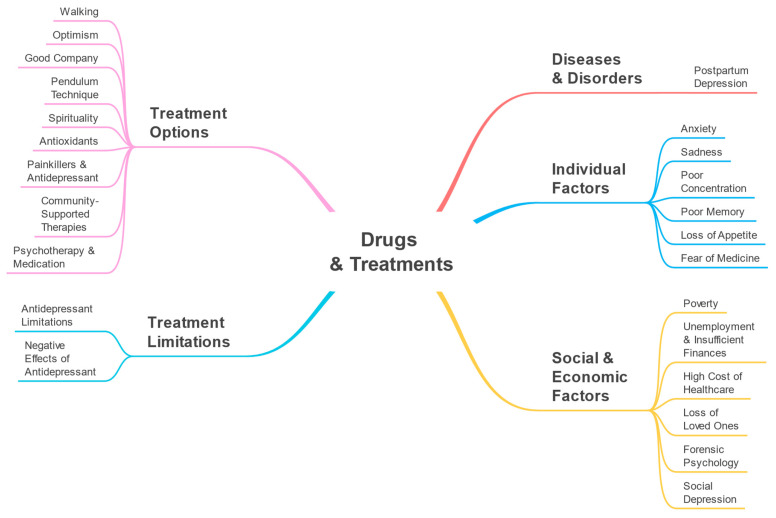
Taxonomy (perspective: Drugs and Treatments).

**Figure 3 toxics-11-00287-f003:**
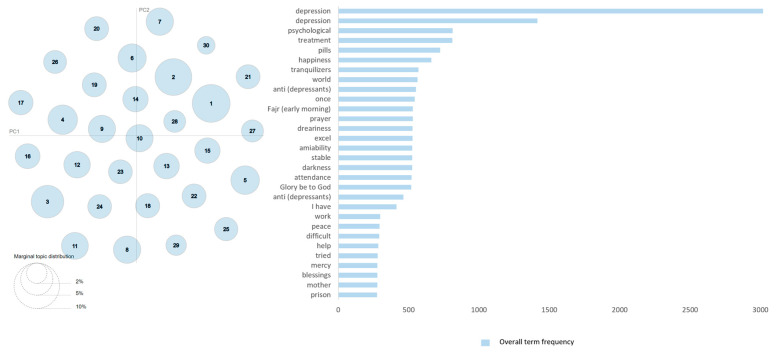
The intertopic distance map of the parameters.

**Figure 4 toxics-11-00287-f004:**
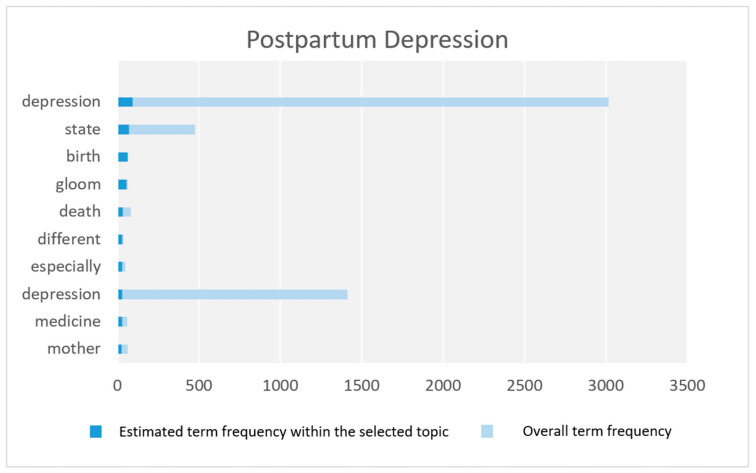
Keyword frequencies (macro-parameter: Diseases and Disorders, perspective: Drugs and Treatments).

**Figure 5 toxics-11-00287-f005:**
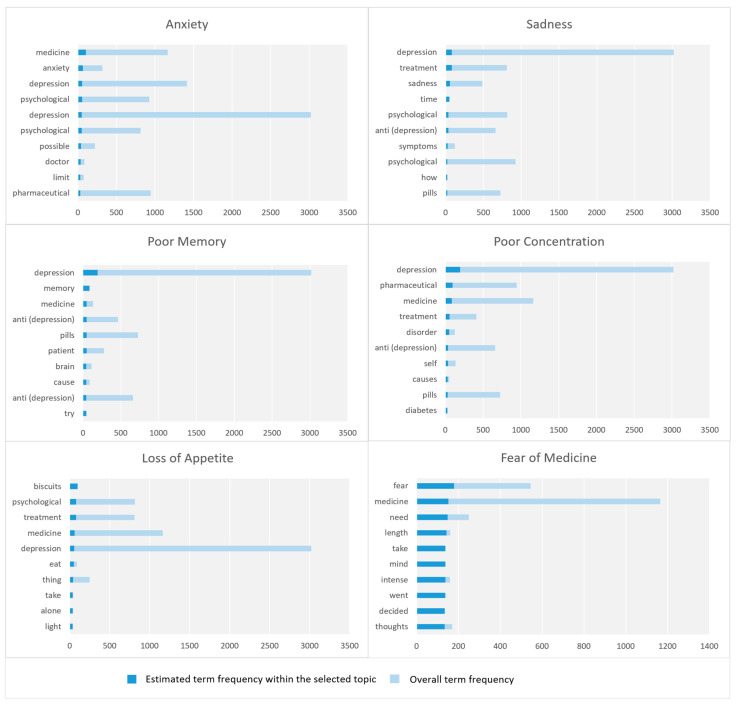
Keyword frequencies (macro-parameter: Individual Factors, perspective: Drugs and Treatments).

**Figure 6 toxics-11-00287-f006:**
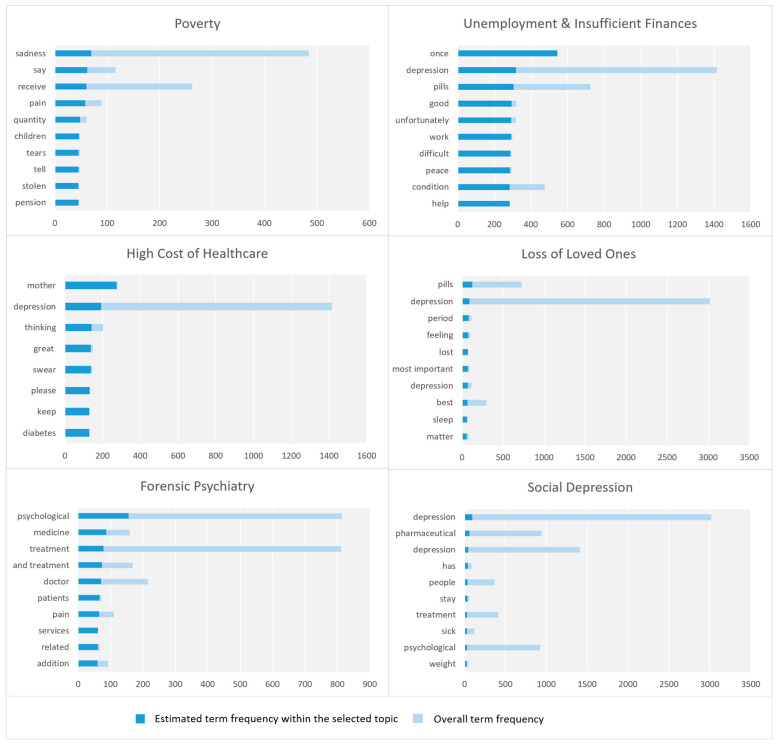
Keyword frequencies (macro-parameter: Social and Economic Factors, perspective: Drugs and Treatments).

**Figure 7 toxics-11-00287-f007:**
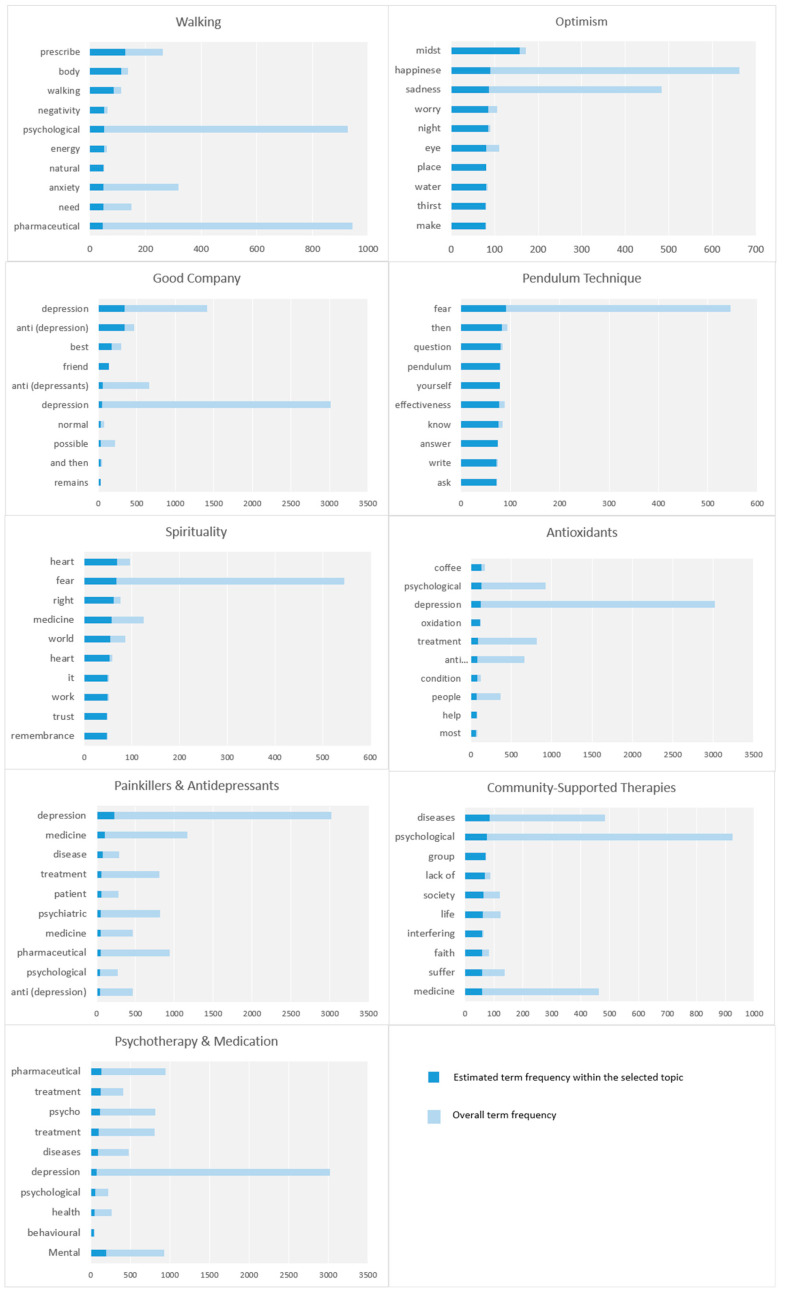
Keyword frequencies (macro-parameter: Treatment Options, perspective: Drugs and Treatments).

**Figure 8 toxics-11-00287-f008:**
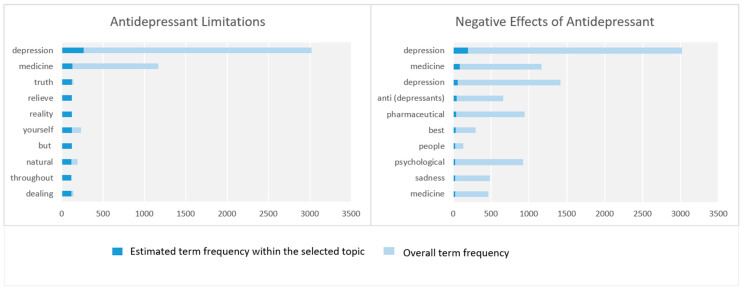
Keyword frequencies (macro-parameter: Treatment Limitations, perspective: Drugs and Treatments).

**Figure 9 toxics-11-00287-f009:**
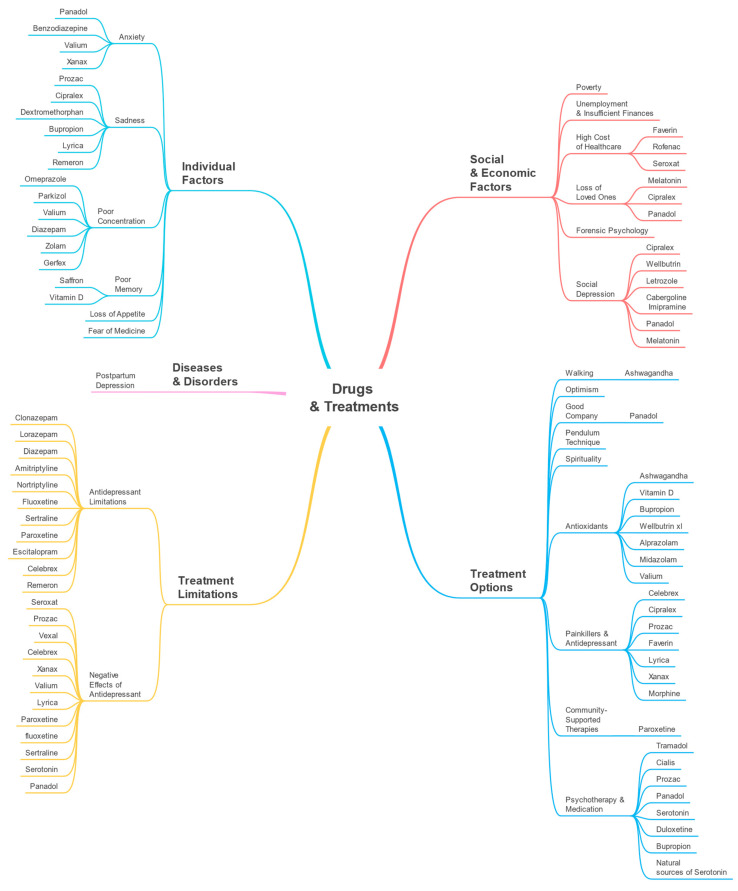
Parameter-drug associations maps (perspective: Drugs and Treatments).

**Figure 10 toxics-11-00287-f010:**
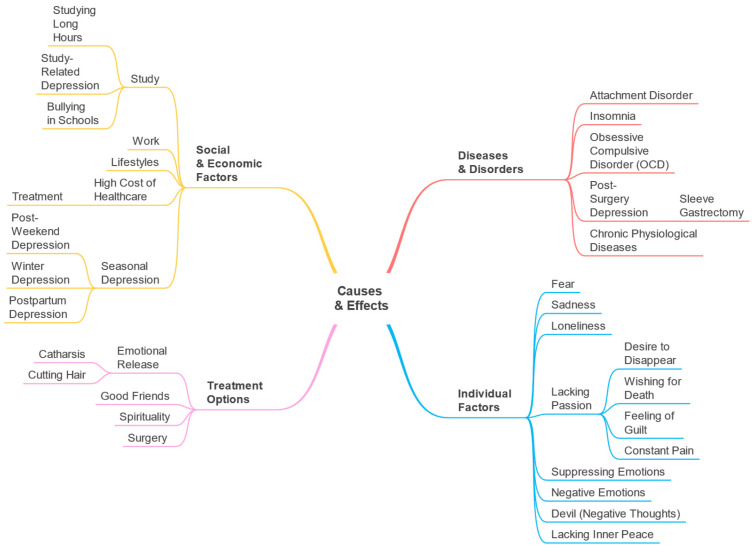
Taxonomy (perspective: Causes and Effects).

**Figure 11 toxics-11-00287-f011:**
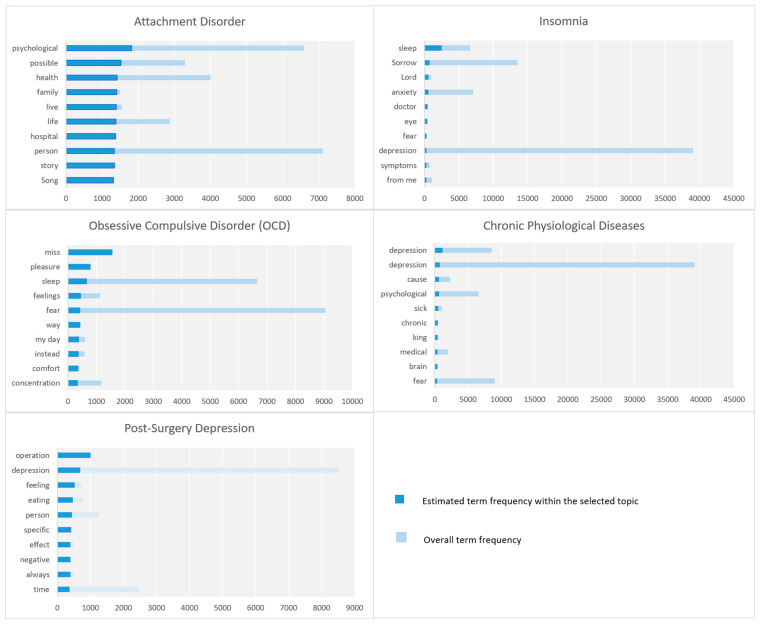
Keyword frequencies (macro-parameter: Diseases and Disorders, perspective: Causes and Effects).

**Figure 12 toxics-11-00287-f012:**
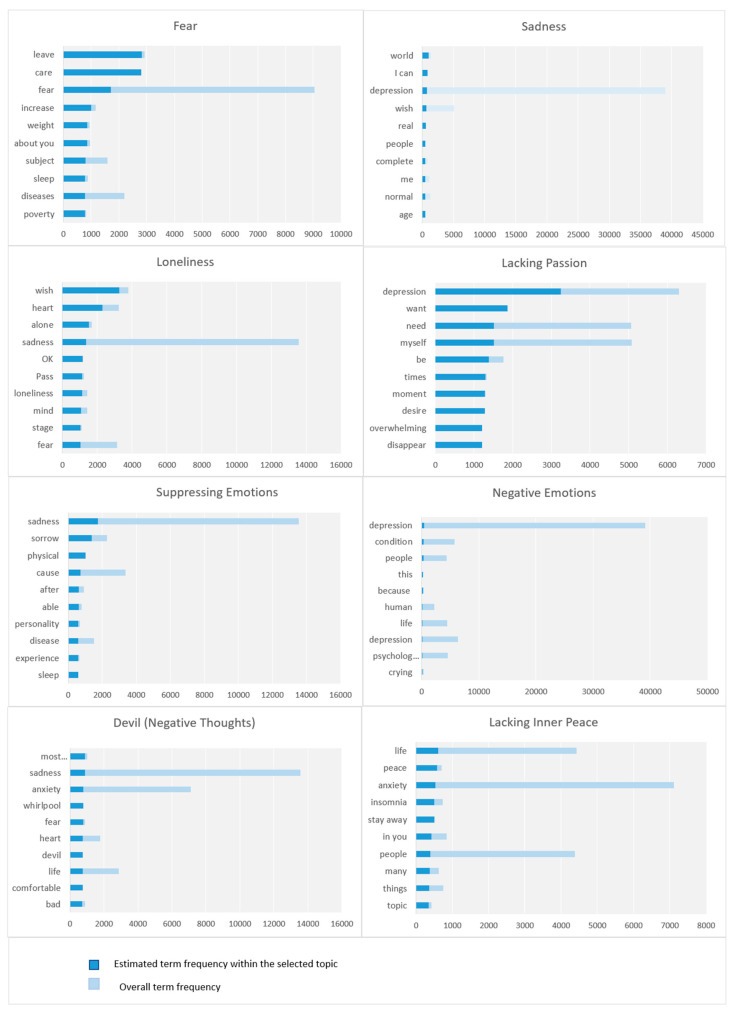
Keyword frequencies (macro-parameter: Individual Factors, perspective: Causes and Effects).

**Figure 13 toxics-11-00287-f013:**
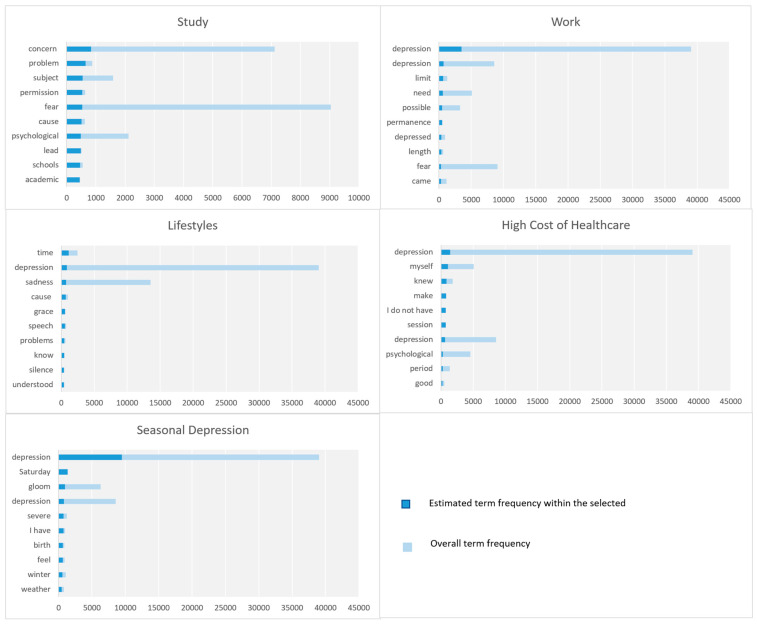
Keyword frequencies (macro-parameter: Social and Economic Factors, perspective: Causes and Effects).

**Figure 14 toxics-11-00287-f014:**
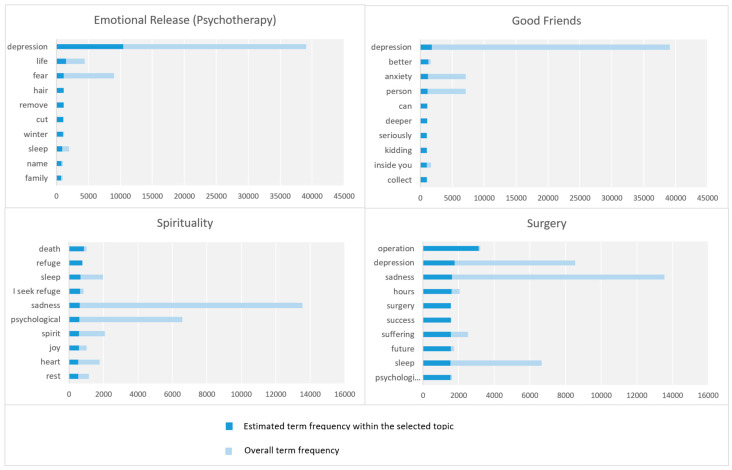
Keyword frequencies (macro-parameter: Treatment Options, perspective: Causes and Effects).

**Figure 15 toxics-11-00287-f015:**
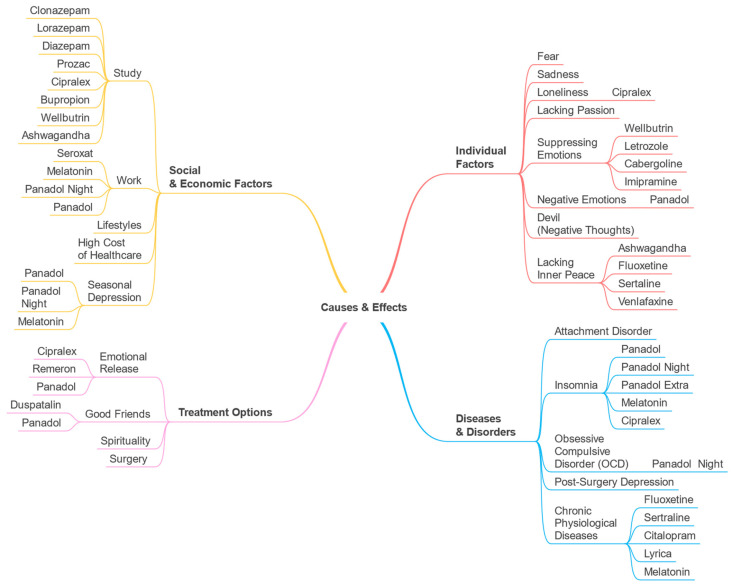
Parameter-drug associations maps (perspective: Causes and Effects).

**Figure 16 toxics-11-00287-f016:**
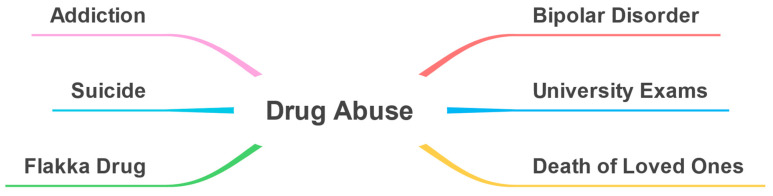
Taxonomy (perspective: Drug Abuse).

**Figure 17 toxics-11-00287-f017:**
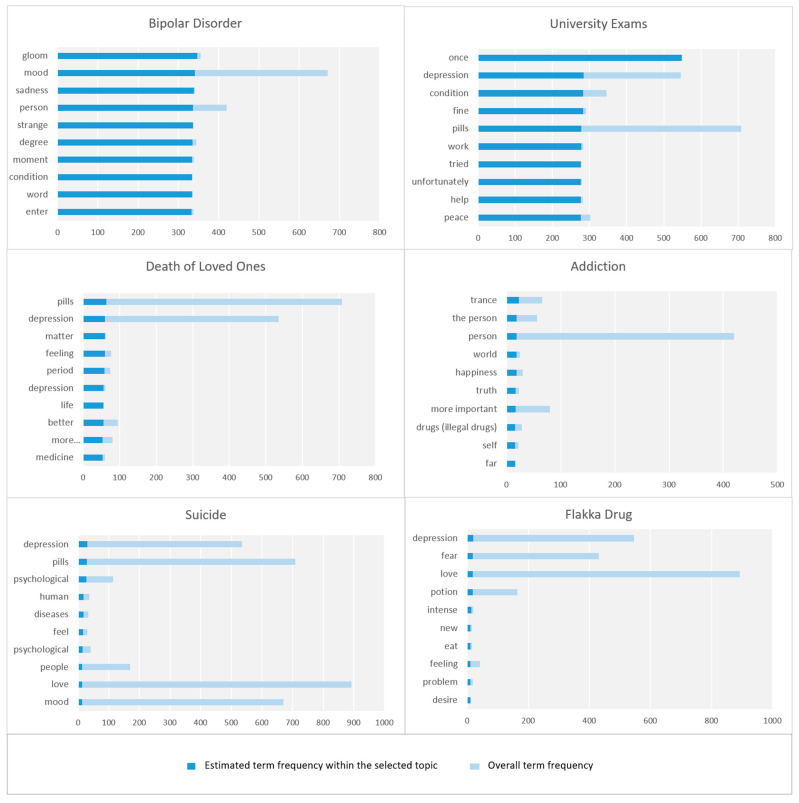
Keyword frequencies (macro-parameter: Drug Abuse, perspective: Drug Abuse).

**Figure 18 toxics-11-00287-f018:**
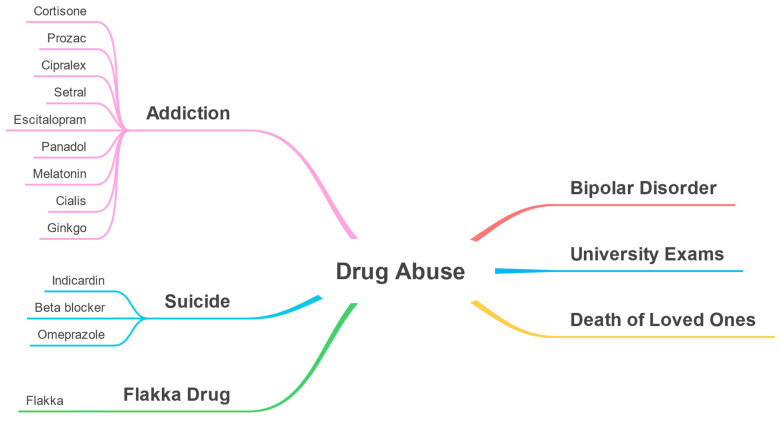
Parameter-drug associations maps (perspective: Drug Abuse).

**Figure 19 toxics-11-00287-f019:**
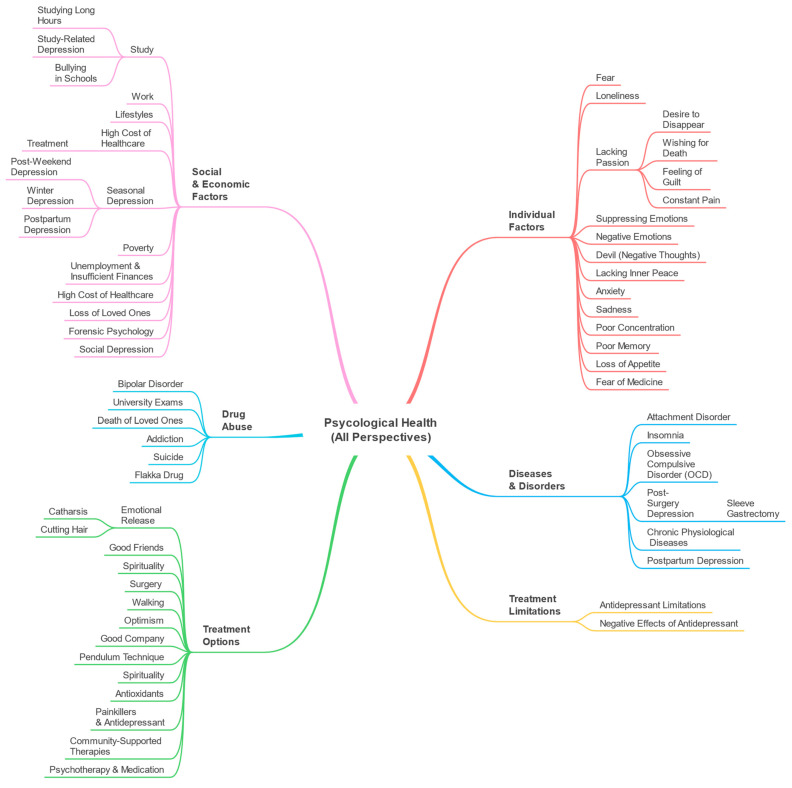
Taxonomy (perspectives: Drugs and Treatments, Causes and Effects, and Drug Abuse).

**Table 1 toxics-11-00287-t001:** Macro-parameters and parameters (perspective: Drugs and Treatments).

Macro-Parameter	Parameters	No.	(%)	Keywords
Diseases and Disorders	Postpartum Depression	29	2	depression, state, birth, gloom, death, different, especially, depression, medicine, mother, afflict, women, usually, sadness, husband, advise, hate, postpartum, call, first
Individual Factors	Anxiety	14	3.1	medicine, anxiety, depression, psychological, depression, psychological, possible, doctor, limit, pharmaceutical, blessing, obsessive-compulsive disorder, sleep, pain, treatment, great, health, book, dose
	Sadness	18	2.8	depression, treatment, sadness, time, psychological, anti (depression), symptoms, psychological, how, pills, treatment, wonder, deep, disappointed, hopes, wound, in, re-in, psychiatric, heal
	Poor Concentration	19	2.8	depression, pharmaceutical, medicine, treatment, disorder, anti (depression), self, causes, pills, diabetes, a lot, deficiency, anxiety, prescription, diseases, praise be to God, treatment, psychological, depression, dangerous
	Poor Memory	10	3.6	depression, memory, medicine, anti (depression), pills, patient, brain, cause, anti (depression), try, because, others, weakness, concentration, important, cause, dangerous, that, unknowingly, effects
	Loss of Appetite	27	2.4	biscuits, psychological, treatment, medicine, depression, eat, thing, take, alone, light, first, sat, in, number, coffee, chocolate, food, great, dispute, side
	Fear of Medicine	3	5.1	fear, medicine, need, length, take, mind, intense, went, decided, thoughts, feelings, now, have, take, no, help, feelings, wellness, end
Social and Economic Factors	Poverty	26	2.5	sadness, say, receive, pain, quantity, children, tears, tell, pension, stolen, waiting, bear, medicines, diseases, depression, psychological, fear, dwelling, strong, psychological
	Unemployment and Insufficient Finances	2	6.5	once, depression, pills, good, unfortunately, work, difficult, peace, condition, help, tired, tried, mercy, blessings, prison, sons and daughters, suicide, bring, have, seeker
	High Cost of Healthcare	4	4.2	mother, depression, thinking, swear, great, please, keep, diabetes, pay, pay her medication, incapacitated, sleep, electricity, income, sick, tightness, elderly widow, cheer, hypertension, bill
	Loss of Loved Ones	21	2.8	pills, depression, period, feeling, lost, most important, depression, best, sleep, matter, medicine, even, life, living, death, friend, desire, I, Iniesta, wife
	Forensic Psychiatry	24	2.7	psychiatry, medicine, treatment, and treatment, doctor, patients, pain, services, related, addition, knowledge, provision, efficiency, facilitation, medication, interaction, pertaining, trial, including, specifically
	Social Depression	22	2.8	depression, pharmaceutical, depression, has, people, stay, treatment, sick, psychological, weight, take, treatment, medication, anti (depression), bigger, life, city, increase, when, stress
		25	2.6	
Treatment Options	Walking	15	3.1	prescribe, body, walking, negativity, psychological, energy, nature, anxiety, needs, pharmaceutical, diseases, fear, equivalent, work, painkillers, emptying, endorphins, sedatives, secrete, reduce
	Optimism	17	2.9	midst, happiness, sadness, worry, night, eye, place, water, thirst, make, cross, thunder, blackness, bridge, darkness, sight, make, grow, chagrin, whiteness
	Good Company	16	3	depression, anti (depression), best, friend, anti (depressants), normal, possible, and then, remains, do, good, small, defect, floor, fifth, job, take, remains, introductions
	Pendulum Technique	28	2.3	fear, then, question, pendulum, yourself, effectiveness, know, answer, write, ask, feelings, attachment, ready, mention, answer, sharp, intention, depression, anti (depressants), sun
	Spirituality	1	6.9	heart, fear, right, medicine, world, heart, it, work, trust, remembrance, goodness, womb, infiltrate, cut off, cheap, boredom, affliction, depression, and as long as, stream
		23	2.7	
		30	1.5	
	Antioxidants	11	3.5	coffee, psychological, depression, oxidation, treatment, anti (depression), condition, people, helps, most, moods, relieve, improve, simple, anti (oxidants), richness, fruits, combined, plus, vegetables
	Painkillers and Antidepressants	7	3.6	depression, medicine, disease, treatment, patient, psychiatric, medication, pharmaceutical, psychological, anti (depression), instead of, doctor, depression, for a patient, Cipralex, painkiller, body, give, Celebrex, hurt
	Community-Supported Therapies	9	3.6	diseases, psychological, group, lack of, society, life, interfering, faith, suffer, medicine, stigma, factors, the factors, deficiency, hereditary, healthy, therefore, requires, support, sport
	Psychotherapy and Medication	6	3.9	psychiatric, pharmaceutical, treatment, psychiatric, treatment, diseases, depression, psychological, health, behavioral, drugs, doctor, psychiatric, medicinal, medicine, disease, drug, illness, pharmaceutical, psychiatrists
		13	3	
Treatment Limitations	Antidepressant Limitations	5	4	depression, medicine, truth, relieve, reality, yourself, but, natural, throughout, dealing, mind, so, crises, those, right, exaggerating, delight, emotion, happiness, nervousness, help
	Negative Effects of Antidepressant	8	3.6	depression, medicine, depression, anti (depressants), medicines, best, people, psychological, sadness, medicine, possible, pill, condition, psychological, actually, disease, diseases, there is, nervousness, causes
		20	2.8	
		12	3.4	

**Table 2 toxics-11-00287-t002:** Macro-parameters and parameters (perspective: Causes and Effects).

Macro-Parameter	Parameter	No.	(%)	Keywords
Diseases and Disorders	Attachment Disorder	8	3.8	psychological, possible, health, family, live, your life, hospital, person, story, song, reality, well-being, success, locked up, lost, attachment, audience, money, sung by her
	Insomnia	12	3.2	sleep, sadness, Lord, anxiety, doctor, eye, fear, depression, symptoms, from me, I am, fear, name, diaspora, myself, when, teach, blessings, matter
		24	2.5	
	Obsessive Compulsive Disorder (OCD)	30	2.2	miss, pleasure, sleep, feeling, fear, way, daily, instead, comfort, concentration, my life, depression, thinking, habit, calm, depression, self, review, mental, practice
	Post-Surgery Depression	23	2.6	operation, depression, feeling, eating, person, specific, effect, negative, always, time, stomach, eat, medical, happen, food, support, loneliness, for you, eat, get out
	Chronic physiological Diseases	9	3.7	depression, depression, cause, psychological, sick, chronic, king, medical, brain, fear, diseases, Salman, suffering, surgical, cause, city, relationship, psychological, nerves, compensate
Individual Factors	Fear	16	2.9	leave, care, fear, increase, weight, about you, subject, sleep, diseases, poverty, keep away, think, and so on, difference, fear, doctor, health, face, your fear, sources
	Sadness	19	2.9	world, I can, depression, wish, real, people, complete, me, normal, age, try, need, needs, work, fear, person, I, years, time, stay
	Loneliness	4	4.6	wish, heart, alone, sadness, ok, pass, loneliness, mind, stage, fear, focus, nights, human, thinking, anxiety, unknown, details, compensate, trust, calm down
	Lacking Passion	11	3.3	depression, want, need, myself, be, times, moment, desire, overwhelming, disappear, the world, have, presence, heavy, exist, feel, want, depression, sadness, view
		15	2.9	
	Suppressing Emotions	17	2.9	sadness, sorrow, physical, cause, after, able, personality, disease, experience, sleep, possible, upset/angry, need, your chest, was not, wish, tell, say, inside, live
	Negative Emotions	21	2.7	depression, condition, people, this, because, human, life, depression, psychological, crying, sleep, conversation, life, yourself, have, sadness, anxiety, permanent, phrase, love
	Devil (Negative Thoughts)	22	2.7	most important, sadness, anxiety, whirlpool, fear, heart, devil, life, comfortable, bad, sorrows, stable, caused, current, last, past, make, tense, destroy, cultivate
	Lacking Inner Peace	29	2.2	life, peace, anxiety, insomnia, stay away, in you, people, many, things, topic, anger, inside me, focus, your Lord, struggle, fear, anxiety, psychological, joy
Social and Economic Factors	Study	6	4.2	concern, problem, subject, permission, fear, cause, psychological, lead, schools, academic, level, impact, delay, space, going, coming, elite, to school, disability, counsellors
		7	4	
		14	2.9	
	Work	5	4.5	depression, limit, need, possible, permanence, depressed, length, fear, came, no one, praise be to God, literally, still, life, sufficiency, society, psychological, coming, deficiency
	Lifestyles	25	2.5	time, depression, sadness, cause, grace, speech, problems, know, silence, understood, inside, pretended, stupid, committed, smiled, answered, wellness, weight, in relation to, hospital
	High Cost of Healthcare	26	2.4	depression, myself, knew, make, I don’t have, session, depression, psychological, period, good, for depression, seasons, diseases, suffering, fear, difficult, home, street, family, life
	Seasonal Depression	2	5.7	depression, Saturday, gloom, depression, severe, I have, birth, feel, winter, weather, spray, period, know, month, offender, people, cause, feel, atmosphere, inside
		18	2.9%	
Treatment Options	Emotional Release (Psychotherapy)	1	6.3	depression, life, fear, hair, remove, cut, winter, sleep, name, family, wake up, satiate, inside, side, entered, smell, bring, come, answer, people
	Good Friends	10	3.4	depression, better, anxiety, person, can, deeper, seriously, kidding, inside you, collect, spontaneity, quest, reach, continuity, wonderful, include you, the two things, the mother, cause, not happened
		13	3	
	Spirituality	20	2.8	death, refuge, sleep, I seek refuge, sadness, psychological, spirit, joy, heart, rest, life, body, soul, society, anxiety, question, injustice, conditions, blackness, break
	Surgery	3	5.6	operation, depression, sadness, hours, surgery, success, suffering, future, sleep, psychological, medical, patient, mood, Salman, natural, first, thinking, anxiety, excess, permanent

**Table 3 toxics-11-00287-t003:** Macro-parameters and parameters (perspective: Drug Abuse).

**Parameter**	**No.**	**(%)**	**Keywords**
Bipolar Disorder	1	16.1	gloom, mood, sadness, person, strange, degree, moment, condition, word, enter, logical, random, transform, waves, endurance, need, tears, moments, reassurance, understanding
University Exams	2	13.9	once, depression, condition, fine, pills, work, tried, unfortunately, help, peace, family, suicide, prison, answer, mercy, cut, tired, have, difficult, see
Death of Loved Ones	6	4.4	pills, depression, matter, feeling, period, depression, life, better, more important, medicine, even, death, lived, lost, sleep, go, desire, resistance, Kharkhi, pillow
Addiction	7	3.6	trance, the person, person, world, happiness, truth, most important, drugs (illegal drugs), self, far, realistic, fact, closer, weakness, close, health, knowledge, narcissist, connection, dots
	8	3.6	
	24	1.5	
Suicide	19	1.7	depression, pills, psychological, human, diseases, feel, psychological, people, love, mood, a lot, disease, depression, cause, illness, brain, excess, addiction, psychological, anxiety
	25	1.5	
	28	1.3	
Flakka Drug	26	1.5	depression, fear, love, potion, intense, new, take, feeling, problem, desire, alone, therefore, withdrawal, dope, lethargy, drug, to withdraw, attempt, symptoms, depression

## Data Availability

The data is acquired from Twitter under their terms and conditions. Anyone can request the same data from Twitter.

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
