# Peer review of "Psychological Health and Drugs: Data-Driven Discovery of Causes, Treatments, Effects, and Abuses"

_toxics, 2023, doi:10.3390/toxics11030287_

Round 1

Reviewer 1 Report (Previous Reviewer 1)

This paper proposes a big data and machine learning-based approach for  the automatic discovery of parameters related to mental health from Twitter data,

In the Authors' opinion,  the methodology of this research can be extended to other diseases and provides a potential for discovering evidence for forensics toxicology from social and digital media.

I would ask the Authors the reasons why -in the revised version of the paper- they deleted reference number 1 related to the introduction (since lines 35 to 37), affirming  "Several factors are contributing globally to declining social sustainability including  people’s health, economic issues, global events such as the COVID-19 pandemic and environmental disasters and increased social division and polarization 1[1]."

- Please consider, white regard to this quotation (lines 54,55): "However, addiction can also contribute to or exacerbate mental health problems, as the use of substances or engagement in certain behaviors can have negative impacts on mental well-being",  the reference by Albano, G.D. et al.,  Toxicological Findings of Self-Poisoning Suicidal Deaths: A Systematic Review by Countries. Toxics 2022, 10, 654. https://doi.org/10.3390/toxics10110654.   

- it is not very clear, almost in my opinion,  the quote related to smoking and drug of abuse in the field of this research, within the title (Psychological Health and Drugs):  "According to the Centers for Disease Control and Prevention (CDC), cigarette smoking causes over 480,000 deaths in the United States annually, with over 40,000 deaths caused  by second-hand smoke. The smoking habit has caused serious health problems for over  16 million Americans 3[3]"

- Please clarify the meaning of "sentiment analysis" in such a scientific field.

- I really appreciated Figure 9. Parameter-Drug Associations Maps (Perspective: Drugs & Treatments) and Table 2. Macro-Parameters and Parameters (Perspective: Causes & Effects) for clarity.

The paper appears interesting and with novel trends for future research in the field of Psychological Health and Drugs.    

Author Response

Reviewer 2 Report (Previous Reviewer 2)

the authors reviewed the manuscript appropriately.

Author Response

Reviewer 3 Report (Previous Reviewer 3)

  The manuscript "Psychological Health and Drugs: Data-Driven Discovery of 2 causes, treatments, effects and abuses" has been significantly improved.   Regards,    

Author Response

This manuscript is a resubmission of an earlier submission. The following is a list of the peer review reports and author responses from that submission.

Round 1

Reviewer 1 Report

Psychological Health and Drugs: Data-Driven Discovery of  Causes, Treatments, Effects, and Abuses. This is interesting research focusing on interdisciplinary aspects of media data and drugs, by using specific tools of investigation  (Keywords Used to Discover Parameters  (Drugs & Treatments Perspective).  

This paper proposes a big data and machine learning-based approach for the automatic discovery of parameters related to mental health from Twitter data.

In the education field, there is research on the prevalence of psychological illnesses among students and academics and the impact of teachers' mental health on students' achievement [12]. Please consider also a recent study  by: Cannizzaro, E.; Lavanco, G.; Castelli, V.; Cirrincione, L.; Di Majo, D.; Martines, F.; Argo, A.; Plescia, F. Alcohol and Nicotine Use among Adolescents: An Observational Study in a Sicilian Cohort of High School Students. Int. J. Environ. Res. Public Health 202219, 6152. 

The authors provide a comprehensive account of mental health, causes, medicines and treatments, mental health and drug effects, and drug abuse, as seen on Twitter, discussed by the public and health professionals. Moreover, the Authors identify their associations with different drugs.

Authors observed: In this study, we provide a variety of visualization methods of the parameters we have discovered. These are intertopic distance maps, taxonomies, as well as keyword frequency diagrams (both cluster-specific and corpus-wide). Please clarify the meaning of "intertopic distance maps" (see also in fig.3) 

Please specify (table 2) why the Authors used some Keywords (since medicine to Omeprazole) and maybe other terms were excluded.  

Reviewer 2 Report

This paper proposes a big data and machine learning approach to automatically discovering mental health metrics from Twitter data. They used Twitter to collect 1,048,575 tweets in Arabic about psychological health in Saudi Arabia. They built a big data machine learning software tool for this job.

The authors say the work will open new directions for social media-based identification of mental health drug use and abuse, as well as other micro and macro factors related to mental health. The methodology can be extended to other diseases and provides potential for uncovering evidence for forensic toxicology from social and digital media.

The work is well written and designed. it is certainly a novelty but I don't think it falls within the target of the magazine. The conclusions reached by the authors are not confirmed by scientific data or by objectivity. How did the authors demonstrate that their results are scientifically valid? Furthermore, the manuscript is excessively long and in some aspects difficult to understand.
